# Agency within Neighborhoods: Multi-Scalar Relations between Urban Form and Social Actors

Ilaria Geddes [1,*], Christakis Chatzichristou [2], Nadia Charalambous [1] and Ana Ricchiardi [1]

1 Society and Urban Form Research Lab., Department of Architecture, University of Cyprus, Nicosia 1678, Cyprus; charalambous.nadia@ucy.ac.cy (N.C.); aricch01@ucy.ac.cy (A.R.)
2 Department of Architecture, University of Cyprus, Nicosia 1678, Cyprus; hadjichristos@ucy.ac.cy
* Correspondence: ilaria.geddes@gmail.com; Tel.: +357-96368308

**Abstract:** This research provides an abstract representation of neighborhoods, accounting for the actors involved in the process of their formation and transformation as local entities embedded in a complex yet specific configuration of historical, social, structural, and political contexts. The analysis uses a conceptual framework combining Assemblage Theory and Actor–Network Theory to examine how both human and non-human actors or agents interact and consequently exert an impact on three different neighborhoods in Limassol, Cyprus. The methodology combines both qualitative as well as quantitative approaches. The tools used include space syntax, land use, and building typology, descriptive statistics of social factors, a photographic survey, observation of the built environment's expressive features, and archival research of press articles. The findings reveal the extent to which global factors, such as colonialism and the mobility of wealthy populations from former Soviet countries, have an impact on the social makeup and expressive features of the environment, while local factors, such as block size and housing typology, have a strong impact on the use of public space. Furthermore, more complex networks may exhibit structural resilience or adaptability but may be, at the same time, more sensitive to varying and conflicting interests.

**Keywords:** agency; Assemblage Theory; Actor–Network Theory; urban form; space syntax; mixed methods





## 1. Introduction

This paper explores urban neighborhoods as dynamic ecosystems with multi-scalar relationships within themselves and with the whole city. Neighborhoods, much like cities, continuously evolve in terms of their physical structures, social compositions, street layouts, public spaces, and the way their residents utilize and inhabit them. However, our attempts to define and describe neighborhoods often oversimplify their complex social networks. This tends to happen to streamline analysis and may be partially determined by normative biases.

These challenges are magnified in contemporary cities, which exhibit new forms of inclusion and exclusion [1]. The urban fabric, once dense and continuous, has now shifted towards a more diffused, fragmented, open, and atomized form [2]. Additionally, the socio-economic diversification of cities may accentuate, intensify, or challenge critical aspects of the urban landscape and coexistence patterns among communities [3]. This phenomenon is particularly pronounced in Southern European cities, experiencing rapid physical and social transformations due to migration flows and the temporary settlement of transient groups. Such shifts are not new or peculiar in Mediterranean port cities, which have served as hubs for cultural and economic exchange, as well as the movement of people across borders and continents for centuries [4].

Examining the urban form's complexity is hindered by the historical isolation of analytical approaches developed in different countries and research contexts. These approaches have spawned distinct schools of thought, typified by the configurational, the

historico-geographical, the process typological, and the spatial analytical approaches, each associated with specific research centers or individuals. However, recent initiatives have sought to bridge these disparate approaches [5–7] to establish a unified framework for multidisciplinary analysis. Yet, their integration remains underexplored, compounded by the challenges of interpreting centuries of urbanization and human intervention in it.

This paper adopts a relational approach informed by Assemblage Theory developed by DeLanda [8] and Actor-Network Theory (ANT) developed by Bruno Latour [9]. This framework offers a comprehensive approach encompassing the multifaceted elements and processes contributing to the emergence and evolution of the urban form, as well as expressive facets of its physical attributes. By applying a multidisciplinary methodology within a philosophical framework, this approach aims to elucidate how neighborhoods shape their identities and respond to external pressures as cities expand and history unfolds around them; while urbanization is a global process, places respond and adapt to it depending on local conditions [10].

This paper underscores the need to enhance research on the interplay between physical elements of places and the evolving social dynamics within urban environments. It emphasizes the need to consider and analyze the reciprocal impacts of the material and social realms within the domain of urban morphology. Such an approach emphasizes how socio-spatial phenomena are manifested through urban dynamics [11], reflecting relationships between local realities (the neighborhood within the city) and global socio-economic and political forces [12,13]. The goal is to provide, through network diagrams, an abstract representation of neighborhoods and their historical development by accounting for the agency of different actors, including human (individuals and organizations) and non-human actors, physical characteristics, processes, and their interconnections. This research is, therefore, inductive as it addresses a broad research question: how is agency reflected in the physical form and expressive features of neighborhoods? The proposed approach aims to identify the multitude of agencies at play and explore their scale, from global to local, their spheres of influence, and the extent to which they shape the various characteristics exhibited by neighborhoods in relation to the broader cityscape. The objective is to develop an understanding of neighborhoods' complexity and infer the implications of such complexity for the neighborhoods themselves.

*Theoretical Premises*

Neighborhoods, like cities, are both physical and social urban environments [14]. They encompass both tangible material entities: buildings connected by streets, roads, and urban infrastructure [15] and "a system of human activities and interactions" [16] (p. 146). Whether these two dimensions exist independently, with the physical environment merely providing a backdrop to human relations, or whether they are inextricably intertwined and mutually influential remains a subject of ongoing debate within urban studies [17]. It is widely accepted that the organization of space reflects social relationships and that it is a cultural and economic product; this is an idea supported by a range of social theorists, including Simmel [18], Lefebvre [19], and Logan and Molotch [20]. What remains controversial is the extent to which the spatial organization and physical structure of cities impact society, particularly in terms of shaping social behaviors and either reinforcing or diminishing social disparities.

When examining the historical evolution of urban forms, urban morphologists have traditionally emphasized the physical aspects and actors wielding influence over planning decisions [21]. By contrast, sociologists have underscored the significance of group dynamics within the city and the role of everyday routines and social activities in molding a city's identity [16]. This bias toward the physical dimension is particularly evident in various approaches to urban morphology, where the primary focus remains on analyzing the physical form of the city (streets, buildings, plots, areas, lines, etc.). The development of complex systems theory has shifted attention toward exploring the interplay between physical and social aspects: a perspective encapsulated in Batty's statement asserting that

"our concern is still with the physical structure of the cities in terms of their geometry and morphology… the new paradigm emphasizing interactions still requires us to express our understanding and designs in physical terms, often as networks, but networks built on strong and significant socio-economic relationships" [22] (p. 19). However, within urban morphology, the material elements still tend to take precedence, with the social dimension often treated as an adjunct; while all urban morphological approaches can and do incorporate the interrelationships between physical and human features, as well as spatial and temporal relations—as summarized in Table 1—there persists a view that the physical form should serve as a common reference in urban analysis [5].

**Table 1.** Components of the urban form and their relationality in different urban morphological approaches. Source: the authors.

| Approach | Physical Features | Social Features | Spatial Relations | Human–Physical Relations | Temporal Relations |
|---|---|---|---|---|---|
| Historico-geographical | Site<br>Town plan (street, plot, building) | Function<br>Land use pattern | Street pattern<br>Plot pattern<br>Building pattern | Social and economic context | Cyclical change |
| Process typological | Building<br><br>Urban tissue<br>District<br>City | Cultural context<br><br>Historical context | Aggregation | Intention<br><br>Construction | Derivation (cyclical reproduction, modification of form) |
| Configurational | Street<br>Open space | Use<br>Occupation<br>Movement | Network structure<br>Interconnectivity | Perception<br>Movement economy<br>Cultural context | Cyclical growth<br>Diversification |
| Spatial analytical | Plot<br>Parcel<br>Census tract<br>Built up area<br>Route | Use | Network structure | Flows | Feedback (continuous readjustment) |

In this context, the authors argue that an approach primarily centered on physical aspects alone is insufficient for analyzing complex entities that operate at various scales. This limitation arises from the interdependence of physical and social factors, as they "act conjointly to produce significant outcomes" [15] (p. 205). Neighborhoods are shaped by people as well as material elements through the capacities they exert on the environment (agency). Examining the urban form through the lens of agency is relevant because it explains what does or does not happen in an urban environment [21]. A more comprehensive relational framework is proposed to offer new interpretive avenues for unraveling the socio-economic forces, both global and local, that drive neighborhood transformations.

Actor–Network Theory (ANT) and Assemblage Theory raise critical issues concerning the mechanisms behind the emergence and transformation of city forms. Scholars have explored the implications of these theories for geography [23–26], and they have been used to analyze urban development dynamics to capture the relationality of urban systems [27]. Both these relational theories suggest that to develop a research approach capable of addressing the complexities of urban development, it is necessary to conduct the following:

1.  Examine the composition of places by analyzing the relationships between material and human components within places and how these exert their influences.
2.  Account for historical processes and how the stabilization and destabilization of the social occur in order to understand persistence, continuity, and change.
3.  Analyze different scales of relationships between parts and the whole and between distant and local agencies.
4.  Investigate how groups are formed and redistributed. Expressive means produced and articulated during the processes of group formation and the stabilization of group identities are key to determining how the social is expressed in their physical form.
5.  Construct a narrative that effectively represents the variety of actors involved.

ANT approaches social research by treating objects as integral elements of social entities, highlighting the agency[1] of material components in the emergence of social phenomena [28]. In ANT, both human and non-human actors are considered equally relevant [29], and the social is seen as a product of associations among these components ("a *type of connection* between things that are not themselves social" [9] (p. 5)) rather than a mere explanatory factor ("the social has never explained anything; the social has to be explained instead" [9] (p. 97)). Within urban morphology, scholars analyze agency to understand the role of humans in shaping the urban landscape. The key concern lies in identifying the actors involved and determining the extent to which they exert an influence [30]. Prominently, within the historico-geographical tradition, urban morphology research has looked into the impacts of agents who directly and indirectly shape the urban form, as well as the interactions between them; specifically, these include landowners, developers, architects, builders, financiers, planners and local policy-makers [31–34]. While the focus has tended to be on human actors and decision-making processes, as Larkham points out [21], the agency of non-human elements in the shaping of urban landscapes has been treated within urban morphology, including from an ANT perspective [35]. This paper adds to this scholarship by shifting the focus from agency as a process to agency as the capacity to exert an influence. This involves initiating, affecting, or changing a process.

Assemblage Theory defines social entities as "wholes whose properties emerge from the interactions between parts" [8] (p. 5) and argues that discrete social entities at any given scale have objective existence and relative autonomy. It demonstrates how social processes occur at scales beyond the micro and the macro levels, identifying their involvement in the emergence of social wholes from interactions among other entities operating at smaller scales.

Assemblage Theory and ANT are relational approaches grounded in philosophy; as such, they are generic and non-specific in analytical terms and provide only a few concepts for empirical research [29].

Previous research by the authors has shown that these approaches can be valuable in inferring causal pathways leading to the physical and social form of cities, particularly regarding the incorporation of broader structural factors [36–39]. This is a dimension that, traditionally, urban morphological approaches have overlooked, initially focusing on physical aspects before incorporating the analyses of local agents and only recently elaborating on how urban morphology might address wider issues [21]. The non-specific nature of Assemblage Theory and ANT can enable the creation of an analytical framework that integrates tools from urban morphology and social sciences, provided that these are aligned with theoretical underpinnings; non-specificity is a characteristic that is advocated as beneficial when making analytical choices [22,40]. Making these approaches applicable to urban morphological analysis requires decisions regarding how to study places while preserving the complexity of large social phenomena based on the five points detailed above.

Determining which physical and human elements to include, how to identify interactions and relationships, and discern the relevance of scales, variables, and historical processes should be guided by existing urban morphological and social approaches as well as the findings of previous empirical studies. The analytical requirements of the theoretical framework should align with relevant morphological approaches and social

research methods, as detailed in Table 2, based on how they address the interplay between material and human elements.

**Table 2.** Summary of approaches and methodological tools used for each analytical requirement. Source: the authors.

| Analytical Requirement | Relevant Approach/Methodology |
|---|---|
| How material and human elements are connected together | Configurational<br>Process typological<br>Spatial analytical |
| Historical process and temporal aspect | Historico-geographical<br>Process typological |
| Different scales and the relationship of part-to-whole | Configurational<br>Spatial analytical |
| How groups are formed and redistributed | Observations<br>Photographic survey<br>Archival research |
| A narrative where the variety of actors is represented | Archival research<br>Description<br>Emplotment |

## 2. Materials and Methods

Three case studies conducted in Limassol, Cyprus, aim to offer a comprehensive and illustrative description of how various factors, both physical and human, as well as their interactions, contribute to the formation of neighborhoods through different synthesizing mechanisms. Limassol, the second largest city in Cyprus and its largest port had a population of 183,555 inhabitants recorded by the census in 2011. Around 80% were Cypriots, 10% EU nationals and 10% third country nationals. The dominant religion in the city is Greek Orthodox, and the five main minorities are Romanian (3.6%), British (3.3%), Greek (3.1%), Russian (2.3%), and Bulgarian (2.3%).

The geographical location of the city and case study neighborhoods are shown in Figure 1.

The selection of these study areas was based on the analysis of social variables from 2011 census data, ensuring a diverse representation of social demographics within the city. The chosen case study areas span different historical periods to highlight the historical processes underlying their development. Furthermore, the selection of the neighbourhoods was informed by conversations with local experts, who highlighted some problematic features in each of them[2]. The three neighborhoods are outlined below:

1.  Arnaout: Situated in the historic heart of the city, Arnaout has a rich history dating back to the Ottoman period. Originally a Turkish Cypriot neighborhood, it is now predominantly inhabited by Greek Cypriot refugees who were allocated Turkish Cypriot properties following the 1974 war. The area also includes Turkish Cypriots who have resettled here, as well as gypsies. Arnaout is known to have poor housing conditions, a high concentration of unskilled laborers, and a significantly high proportion of Cypriot nationals.
2.  Agios Nikolaos: This neighborhood exhibits typical social characteristics, with a substantial middle-class population and a mix of nationalities that align with the city's average demographics. Its history is marked by the establishment of a Greek cemetery in 1865 and the witnessing of the construction of some of Limassol's earliest workers' housing. Residential development in Agios Nikolaos primarily occurred during the 1950s and 1960s.
3.  Dasoudi: Part of the larger Potamos Germasogeia quarter, Dasoudi is a renowned seaside tourism destination in Limassol that is also popular among the locals. While discussions about tourist development began in the late 1960s and early 1970s, it

gained momentum after the 1974 war and continues to thrive. This area boasts a high concentration of higher-income groups and foreign nationals.

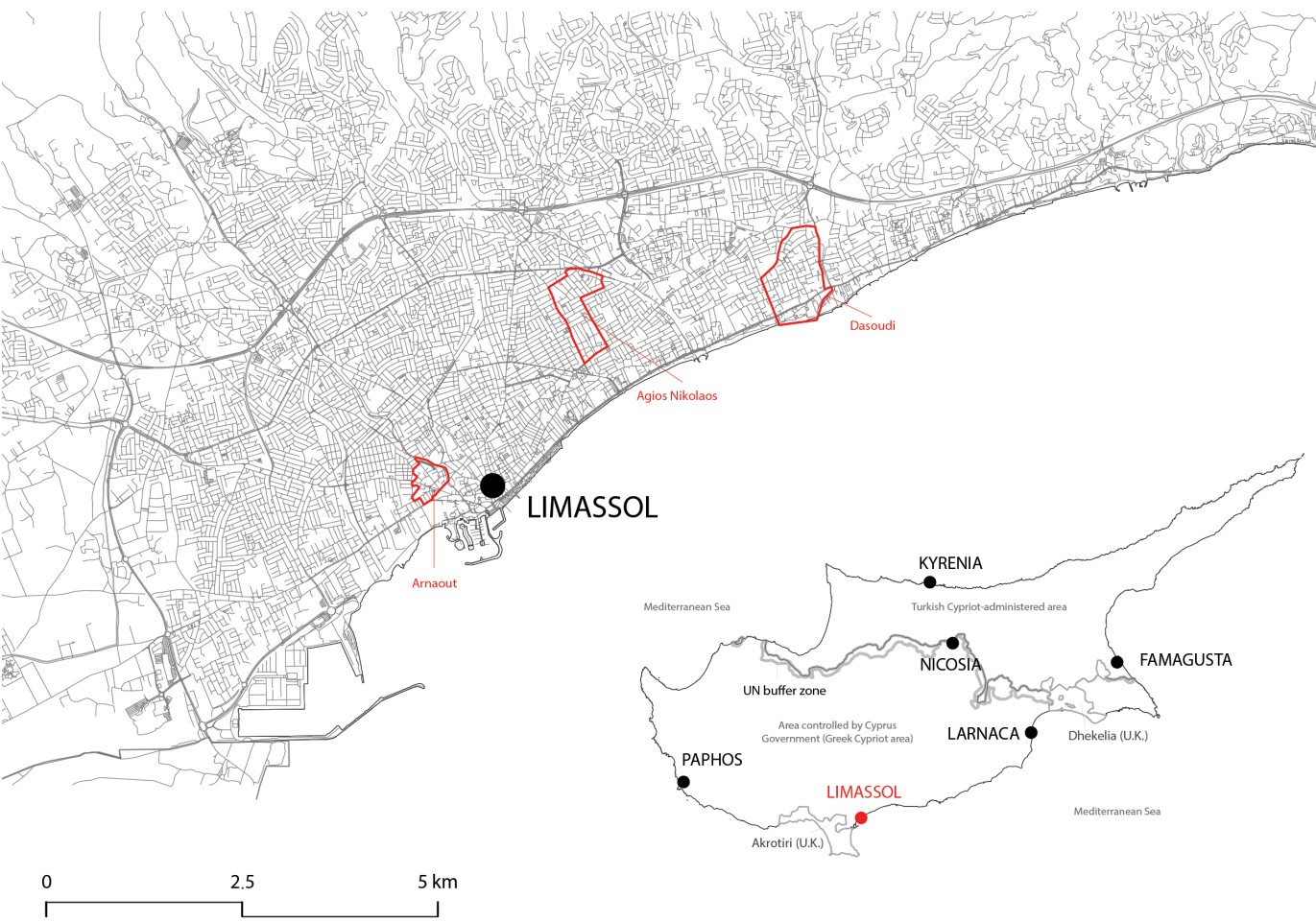

**Figure 1.** Geographical location of Limassol and case study neighborhoods (from west to east: Arnaout, Agios Nikolaos, Dasoudi). Source: the authors. Base map: Bing Maps Satellite Imagery; Inset of Cyprus: adapted from CIA 2010 [41] (Public Domain).

To analyze these neighborhoods, a mixed methodology was employed to perform a multifaceted examination of their development in Limassol and to shed light on the complex interplay between the physical and human factors that have shaped these areas. The methods are summarized below.

The mixed methodology used for the analysis includes the following: (a) a 'spatial history' of the neighborhoods through the mapping of building construction data, space syntax analysis and block size analysis; (b) the construction of a descriptive narrative elaborating on the spatial analyses through archival research of press articles, conversations with three local experts, a researcher's observations and a photographic survey of the sites; and (c) the synthesis of physical and human components, the agencies involved in their development, and their interrelationships in simplified network diagrams. The methodology is summarized in Figure 2, and the methods are explained in more detail below.

### 2.1. Spatial History

This approach involved mapping three layers of information related to the neighborhoods' material components as follows: configurational properties of the street network (space syntax analysis), block size, and building construction dates.

Space syntax[3] quantitatively describes patterns of spatial layouts by analyzing properties of the street network, such as betweenness (NACh—normalized angular choice) and

closeness (NAIn—normalized integration). Betweenness calculates the number of shortest routes passing through a street segment between any two points in the system, reflecting the likelihood of passing movement on the streets. Closeness calculates the relative accessibility of street segments (how close each segment is to all others in terms of the sum of angular changes that are made on each route [44]) within the spatial system, reflecting the likelihood of a street segment to act as an 'easy-to-reach' destination. Measures can be calculated at any scale; for instance, the city-wide scale considering all streets within the city, or local scales considering all streets within a given radius[4].

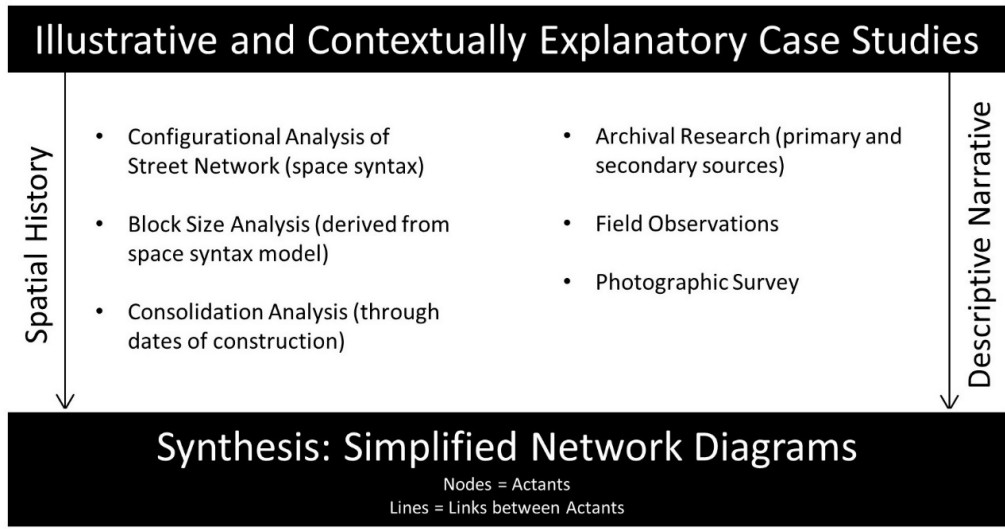

**Figure 2.** Flow chart of research design. Source: the authors.

Block size analysis examines the physical characteristics of city blocks, offering insights into the permeability of the built form, street grid intensification, built density, and compactness of the urban fabric. Block size maps were automatically constructed from the space syntax model of the whole city, and the average block size was calculated for each neighborhood.

The historical consolidation of the neighborhoods was mapped through the construction dates of buildings, helping to visualize their development over time. Dates were digitized as point data from the full archive of the Cyprus Department of Land and Surveys by a private company[5]. The authors integrated the point data into the building footprints in GIS; the data were categorized into specific time intervals corresponding to the periods defined in historical maps (pre-1933, 1934–1960, 1961–1974, 1975–1988, 1989–2003, and post-2003). Historical maps (1883, 1933, 1960, 1974, 1987, 2003) were also used to produce spatial models of the city, facilitating an assessment of the spatial and physical consolidation of the neighborhoods (building construction and block size), as well as assessing the temporal trend of their spatial development.

### 2.2. Descriptive Narrative

The historical context of the city's development was explored through secondary and primary sources, including newspaper archives.

The newspaper archives of the Cyprus Press and Information Office, comprising 38 Greek and English language newspapers dating from 1880 to 2006, and the online archive of the Cyprus Mail were searched for all articles relating to Limassol. The material yielded 10 articles relating to Arnaout and 18 articles relating to Dasoudi. No articles were found relating to Agios Nikolaos.

All areas were visited and observed by a researcher on a Saturday when leisure activities in public areas were likely to be at their peak; in all cases, the weather was sunny and warm[6]. The researcher walked along every street in the area; the photographic

survey involved taking a photograph in each direction from every junction within the area. Additionally, photographs of public green spaces and typical housing types were taken along with other characteristics considered relevant by the researcher based on the preliminary observation of these areas, data from quantitative analyses, and information from conversations. The purpose of the photographic survey was to provide a visual narrative of the neighborhoods to clarify how agency is expressed in their physical form.

*2.3. Synthesis*

After conducting these analyses and drawing inferences, simplified network diagrams were constructed for each neighborhood to represent the interplay between various agencies. In these diagrams, nodes represent actants—anything that has agency in the neighborhood—as follows: an actor, physical or human, but also the resulting characteristics of the urban form, which, in turn, exert influence and contribute to the definition of the identity of a neighborhood. The lines connecting the nodes represent the links between the actants; these are defined as direct links when the 'cause–effect' influence is not immediately intervened upon by another agency or as a mediating link when another agency does intervene. For the purposes of this research, the networks are complete once all the agencies defining the key contemporary characteristics and, consequently, the identity of the neighborhoods, as observed in the analysis, appear in the diagram.

It is important to note that these diagrams are by no means exhaustive; they aim to provide a snapshot of the variety of scales, agencies, and components operating in shaping the identity of the areas.

**3. Results**

The investigation of the case studies integrates various methods and sources to capture the diverse facets of each neighborhood and their interconnections with the entire city. In order to elucidate the causal pathways, the scales of causality, and relational dynamics between different components, a framework that combines Assemblage Theory and Actor-Network Theory (ANT) was employed.

The outcomes of the qualitative analyses were combined with those of the quantitative analyses and descriptive statistics tailored to each specific area. The configurational analysis yielded valuable spatial insights, providing precise measurements and a clear understanding of how each neighborhood is integrated into, and thus influences, the larger urban context.

In this paper, the case studies are organized in chronological order to trace their developmental evolution and contextual significance. This structured approach enables a coherent and in-depth exploration of each neighborhood's unique role within the city.

*3.1. Arnaout*

Arnaout is a neighborhood within the broader Turkish Cypriot area on the western side of the Garyllis River. This neighborhood encompasses several quarters, including Djami Jedit (Tzami Tzatit), which also incorporates much of the medieval core of the city east of the river, including Ayandon (Agios Antonios) and Chiftikler (Tsiflikoudia). Arnout is the northernmost part of this neighborhood, named after its 19th-century settlers of Albanian origin. The settlement occurred during a period when Albania was also under Ottoman rule. The area is shown in Figure 3.

The area comprises several designated public green spaces, including two sections of the newly developed linear park along the Garyllis River and two smaller parks. One of these parks, which also features a small Muslim cemetery, is situated in front of the Arnaout Mosque and the Municipal Administrative Offices responsible for managing Turkish Cypriot properties. These administrative offices have a historical legacy, originally starting out as the People's Home, founded in 1953. Over time, it served various functions, including hosting festivals, balls, wedding ceremonies, conferences, and meetings, and later operating as a hospital until 1974 [47]. The other public green space is on a triangular

piece of land, formerly a Muslim cemetery, located between Misiaouli and Kavazoglou Road, Djelal Bayar Street, and Beyazit Street. Additionally, a small formal public space is embedded within the parking lot of the only purpose-built housing estate in the area.

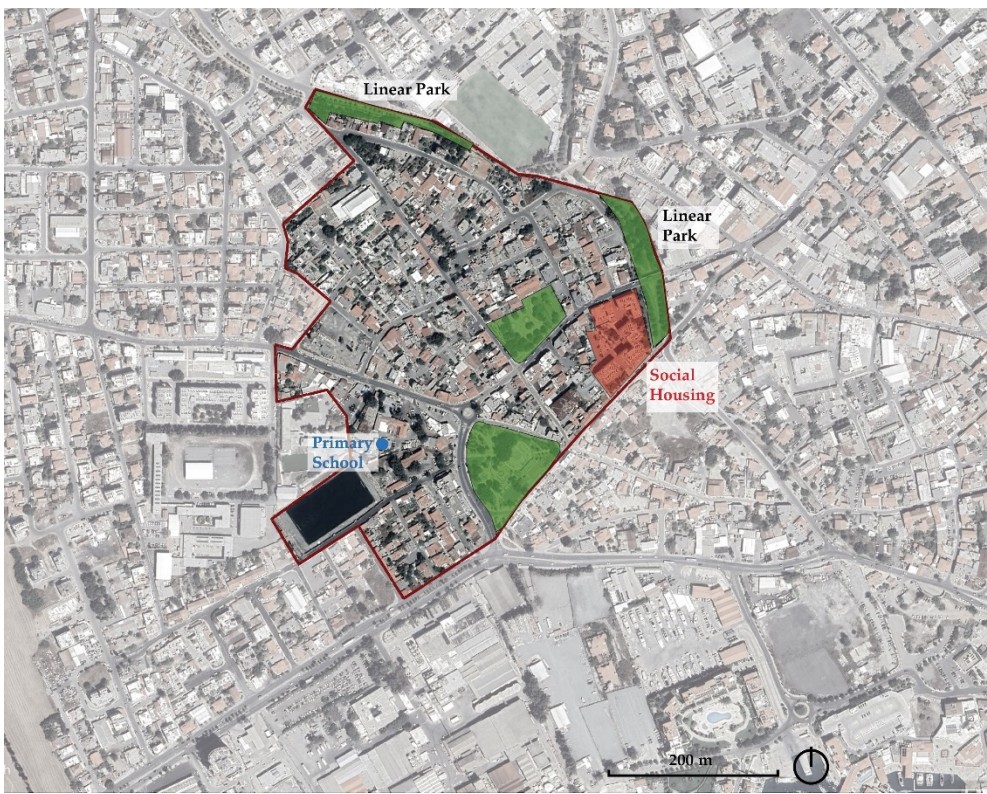

**Figure 3.** The quarter of Arnaout study area. Source: the authors; base map: Google Earth 2023.

During a sunny Saturday morning visit in December, it was noted that none of the public spaces were used, as only one person was seen sitting on a bench in the park in front of the Arnout Mosque. The linear park and triangular gardens appear to be well-maintained (Figure 4).

The gardens in front of the Arnaout Mosque (Figure 5), while relatively clean and well-kept, showed signs of anti-social behavior, including graffiti on benches and the mosque itself, which is not currently in use. Much of the graffiti read 'Saint Antonis', which could be interpreted as a manifestation of ongoing ethnic tension. This might be attributed to younger Greek Cypriots attempting to mark Muslim heritage with the name of the neighboring quarter, Agios Antonios (Saint Anthony).

The public space within the housing estate was in poor condition, with litter lying around and graffiti covering many walls. The overall area in front of the housing estate also lacked maintenance, displaying scattered rubbish, graffiti, and unpleasant odors due to the inadequate cleaning of numerous rubbish bins at the entrance to the estate (Figure 6).

Despite being part of the historical core of the city and characterized by a dense urban fabric, Arnaout has a relatively small population with 905 inhabitants, according to the 2011 census. However, it boasts a high population density of 5256 per square kilometer compared to the city's average of 4221. This density, however, is somewhat lower than in other areas outside the historical center, such as Agios Nikolaos, which is discussed in the following section. This variance is likely due to a significant number of vacant properties (12%) and the predominance of low-rise housing, consisting mostly of one- or two-story buildings, typical of traditional urban areas in Cyprus. Notably, the ground floor of two-story buildings is used for commercial purposes, while the upper floor serves residential purposes.

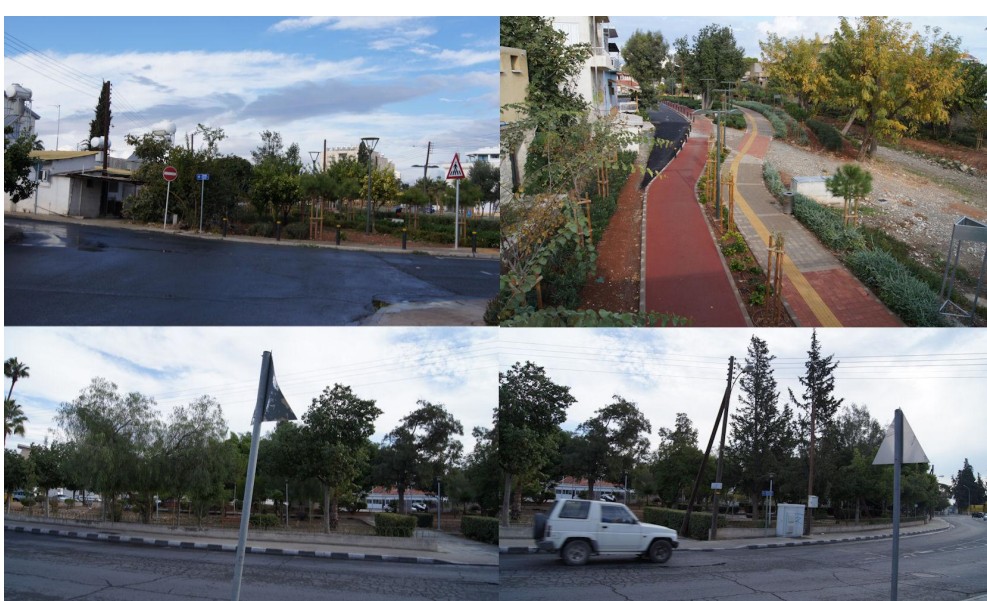

**Figure 4.** Stretches of the linear park (**top**) and the triangular public space along Djelal Bayar (**bottom**). Source: the authors.

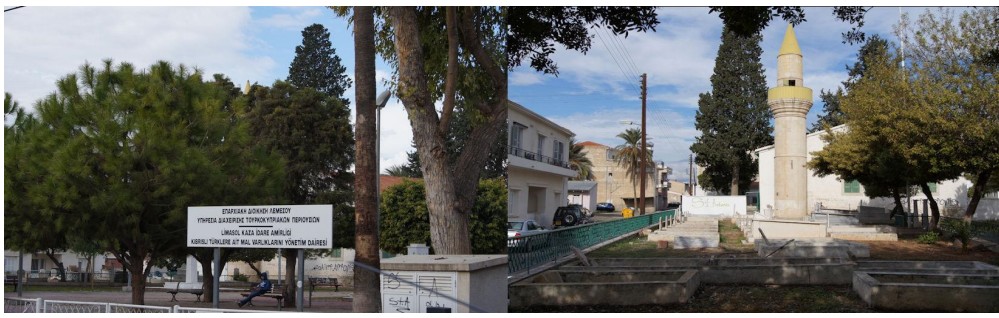

**Figure 5.** Public gardens (**left**) in front of the Arnaout mosque and cemetery (**right**). Source: the authors.

Configurational analysis (Figure 7) revealed a major long-range route running through the southern part of the area (mnNAChN: 1.09), characterized by high vehicular traffic. Several local through-movement routes traverse the area (mnNACh1200: 1.14), also displaying relatively high levels of movement. Despite its central geographical location, the area does not exhibit strong integration at the whole-city scale (mnNAInN: 1.08). Nevertheless, it exhibits excellent local integration, with numerous routes facilitating local movement (mnNAIn1200: 1.23).

The vibrant local activity in Arnaout, characterized by residents walking along main roads and engaging in commercial activities, along with gatherings of male residents in informal public spaces, such as the refugee club near the triangular park and a nearby street, reflects the area's positive local integration values (see Figure 8).

Despite the small population and limited use of public space, the area appears lively, with Greek Cypriots being the primary users. During the observation, a few Southeast Asians were observed entering and exiting properties, though their status as either residents or domestic workers could not be identified conclusively. Additionally, gypsies were seen residing in the area, but no observed interaction occurred between the various groups during the observation. Gypsies were mainly seen engaging in work and recreational activities around their homes, while Southeast Asians appeared to be involved in functional trips. In contracts, Greek Cypriots were actively participating in a variety of activities in public spaces and interacting with one another.

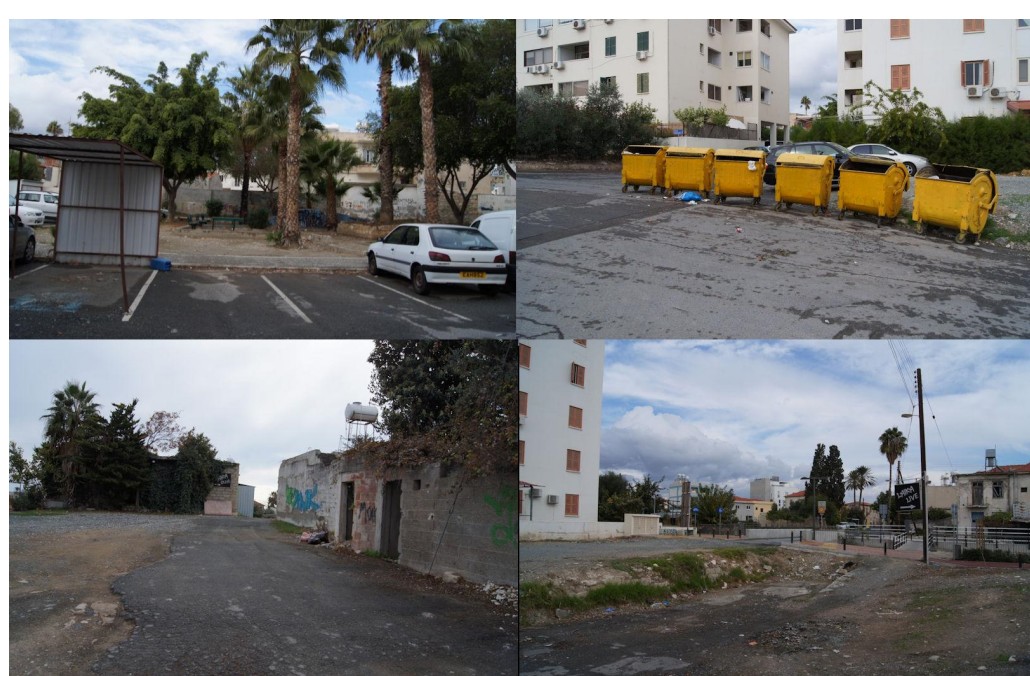

**Figure 6.** Public space within the housing estate (**top left**), refuse facilities in front of the estate (**top right**), and the public realm in front of the estate (**bottom**). Source: the authors.

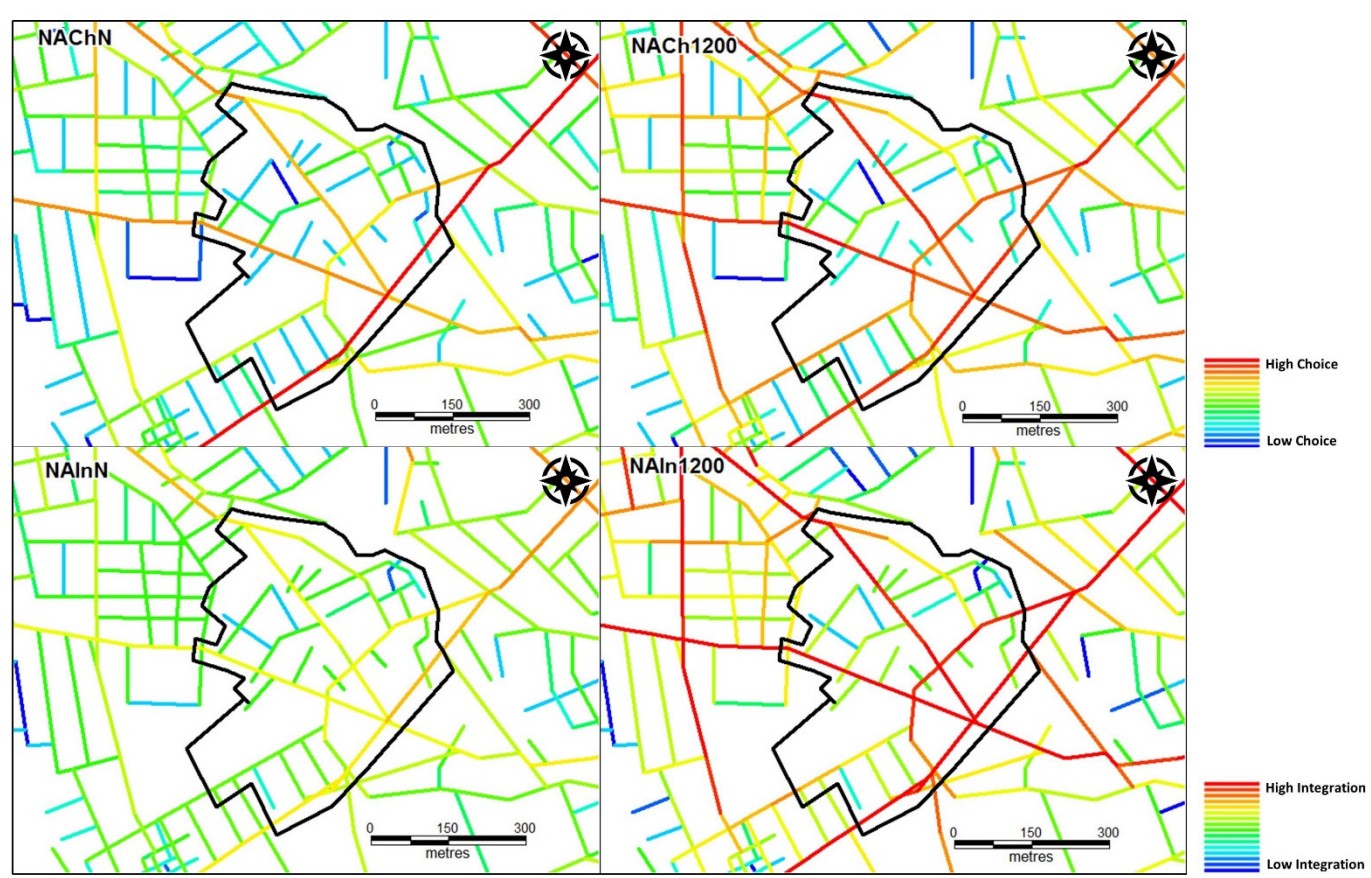

**Figure 7.** Contemporary configurational analysis of Arnaout. Source: the authors.

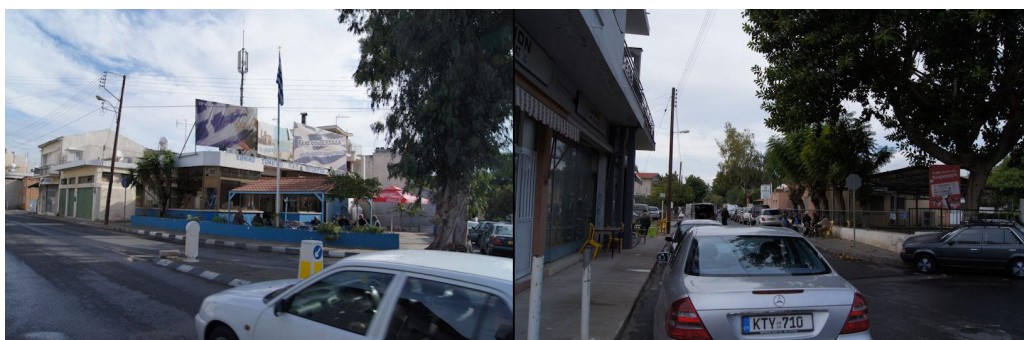

**Figure 8.** The two main areas by the refugee clubs where local residents congregate. Source: the authors.

Corroborating anecdotal accounts and observations, archival newspaper sources provide further insights into the housing and social issues related to this quarter dating back to the early 1990s. These reports highlight the condition of the housing stock, residents' needs for housing improvements, or the search for suitable alternative accommodation. During the observation, many housing units were still in poor condition, particularly those inhabited by gypsies.

Reports from the early 2000s mention disputes over noise and anti-social behavior, particularly in reference to interactions among Greek Cypriots, Turkish Cypriots, and gypsies. Around 70 gypsies moved from the northern part of Cyprus to the southern one and settled in this area following violent discriminatory attacks against them in the early 2000s [48]. The municipality had initially attempted to establish housing for the gypsies on the outskirts of town; however, local residents' protests halted the development. Consequently, the gypsies settled here because of available accommodation at the lower end of the market. Gypsies were known to reside in camps in this area before 1974 [49] and had moved to northern Cyprus along with the Turkish Cypriots after the war.

In the early 2000s, a number of Turkish Cypriots also moved to the area, with a few reclaiming properties they owned while others rented or were accommodated in social housing units. Press reports highlighted issues related to anti-social behavior and fear of crime, disputes, and fights among the various ethnic groups shortly after the arrival of the gypsies and Turkish Cypriots [48]. Social services conducted research on the situation of different ethnic groups in the area and the tensions among them, which was reported in the press [50], indicating that by 30 September 2003, 211 Turkish Cypriots and 360 gypsies had settled in the Djami Jedit and Arnaout quarters. While some Greek Cypriots viewed the newcomers and the social integration programs positively, many others reported concerns such as area degradation, poor cleanliness, a lack of communal spaces, inadequate hygiene, the lower educational level of their neighbors, verbal aggressiveness, dangerous driving, theft, begging and illicit behavior. Turkish Cypriots and gypsies mainly reported problems relating to their living environment and the housing stock, including overcrowded conditions and homes classified as dangerous properties due to their conditions.

The inception of this area traces its roots back to the Ottoman era when the Turkish population settled in the city. The primary catalyst for its emergence can be attributed to the expansive influence of a 'global' agency—exerted by the Ottoman Empire—as it extended its reach to neighboring territories. The establishment of the Turkish Cypriot district transpired through the resettlement of Turkish residents, while the emergence of Arnaout itself was a result of migrations from Albania, although the exact reasons behind these migrations remain unclear.

Critical physical elements that played a pivotal role in shaping the character of this area include the river and its oldest bridges, serving as passages over the narrowest and shallowest points. One bridge is located by the Djami Jedit Mosque, while the other is the 'Four Lanterns Bridge' farther north, constructed by the British in around 1900. The initial lack of connections between the main part of the city and the settlement west of the river resulted in the physical segregation of the area.

As the population grew throughout the latter half of the 19th century and the early 20th century, significant developments, including the establishment of two Muslim cemeteries in the vicinity, the Arnout Mosque, and the conversion of the People's Home into a hospital, contributed to the homogenization of this area. These developments, coupled with the enduring physical segregation, contributed to the area's identity as the Turkish Cypriot district.

The emergence of antagonistic nationalist movements within and outside of Cyprus led to the formation of conflicting groups, escalating into ethnic clashes and, ultimately, the 1974 war. The mediation of this complex situation culminated in the establishment and consolidation of the area's new identity, initially as a Greek Cypriot refugee area and later as a mixed district characterized by a range of problematic aspects encompassing both physical and socio-economic dimensions. The agency of supra-national bodies relates to the segregated location of the area beyond the physical 'boundaries' from the center of town and the typology and quality of its housing stock.

A simplified network diagram illustrating the causal pathways, components, and agencies that contributed to the emergence of this area as a discrete social entity with its own distinctive characteristics is shown in Figure 9.

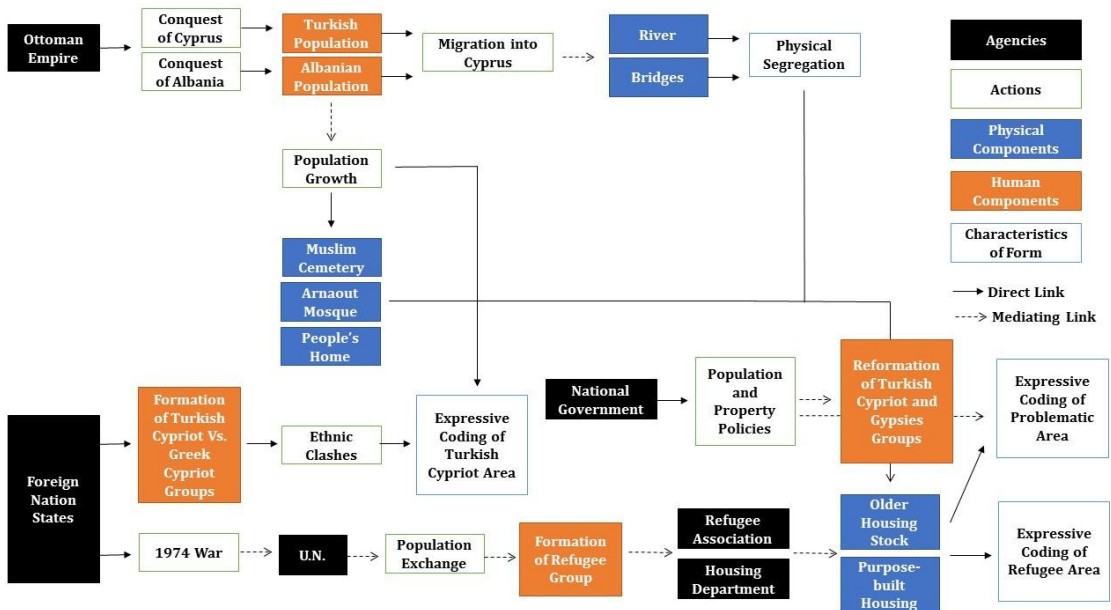

**Figure 9.** Simplified network diagram of Arnaout. Source: the authors.

### 3.2. Agios Nikolaos

Situated in the contemporary heart of Limassol, this area lies just to the northeast of the oldest inner ring road in the city. It encompasses the Greek Orthodox cemetery, established in 1865. Initially modest in size, this cemetery underwent substantial expansion in the aftermath of World War II. The majority of this area falls under the jurisdiction of the Limassol municipality, while its northern segment falls within the administrative boundaries of Mesa Geitonia.

Spanning a considerable area, this neighborhood boasts a relatively high population density, although void spaces are scattered throughout, primarily in the northern part. These void spaces often take the form of empty plots, which are frequently utilized for parking purposes. Two such examples can be observed in Figure 10.

The development of this area commenced in the 1950s, and by 1960, it still retained a significant number of open fields and vacant plots. However, by the year 1974, it had already achieved a population density nearly comparable to its present levels. As of the 2011 census, the area was home to 2508 residents, boasting a notably high population

density of 6506 individuals per square kilometer. This figure stands in contrast to the city-wide average of 4221 despite the presence of the substantially large cemetery block.

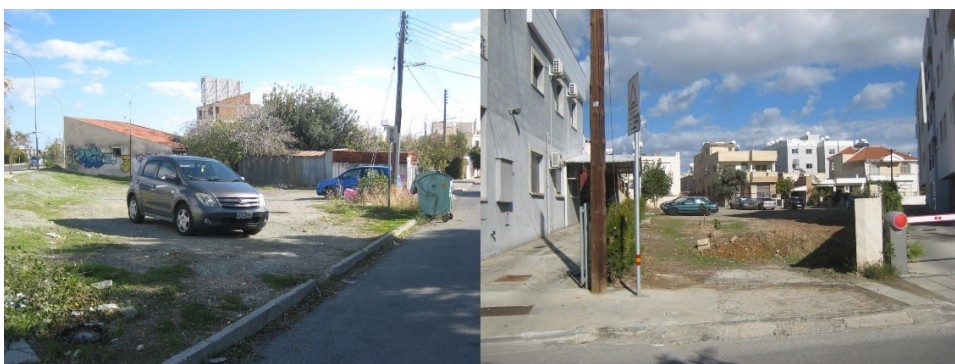

**Figure 10.** Empty plots used for parking within the northern part of the case study area. Source: the authors.

The social analysis underscores that this area predominantly hosts a concentration of middle-class residents. Additionally, the proportion of non-Cypriot citizens and the level of unemployment fall either within the average range (in its northern part) or slightly above average (in its southern part). Further specifics pertaining to this area are illustrated in Figure 11.

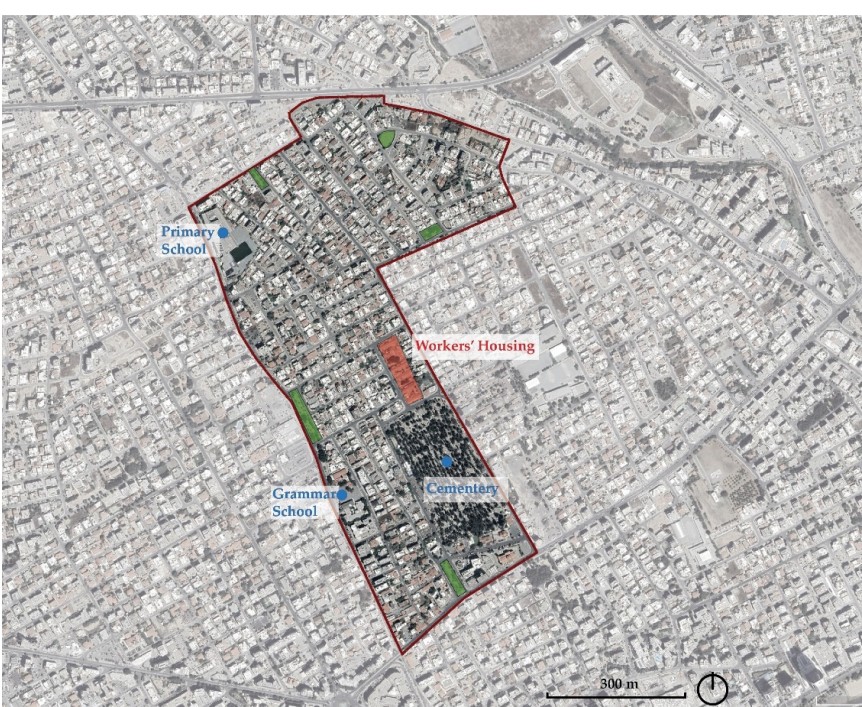

**Figure 11.** The Agios Nikolaos study area. Source: the authors; base map: Google Earth 2023.

Within this neighborhood, there are five designated green spaces, as indicated in Figure 11. These are all well-maintained and include playground facilities. However, during the researcher's visit, only one of these spaces showed any signs of use, with a solitary pedestrian passing through. An 'informal' green space, adorned with aged household chairs and recliners, was identified within a landscaped area between two rows of workers' homes, although it, too, remained unused during the visit. Examples of these spaces are shown in Figure 12.

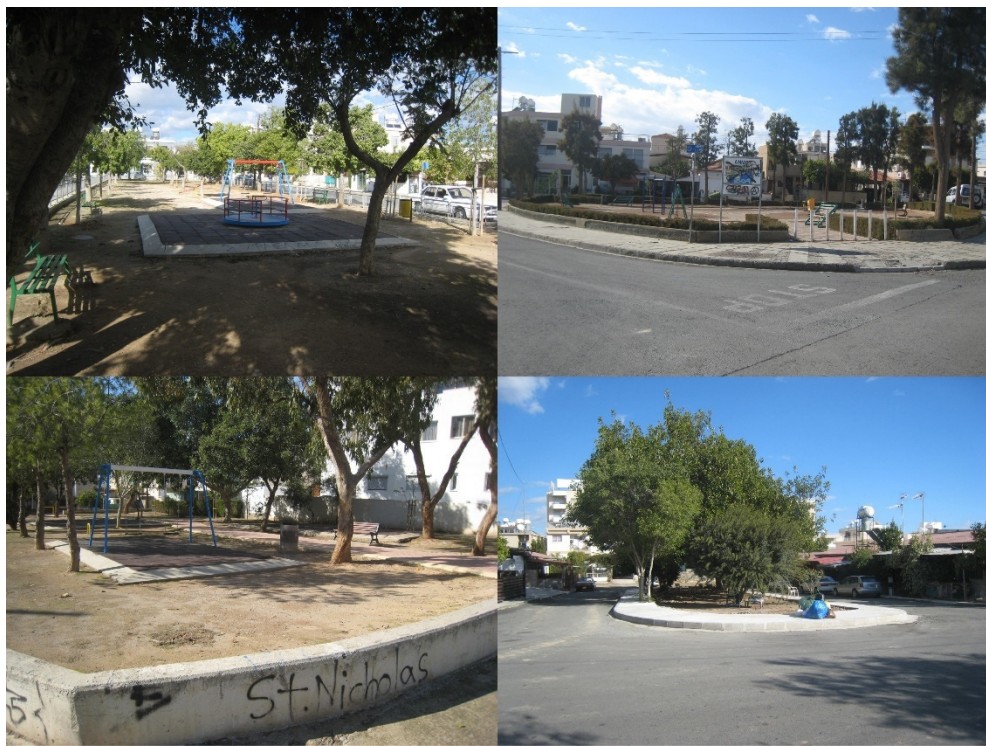

**Figure 12.** Examples of green spaces with playgrounds in the case study area, and the 'informal' space between rows of workers' homes (**bottom right**). Source: the authors.

Throughout the area, a sense of tranquility prevailed, both in terms of pedestrian activity and vehicular traffic. Although a few pedestrians and motorists were noted in the more residential zones, their numbers were relatively low. The primary hubs of movement and commercial establishments were concentrated along the main thoroughfares encompassing the neighborhood, particularly along the roads corresponding to the boundaries of the areas in the south and the west.

Housing in the area primarily consists of small, detached homes or modest apartment complexes, with the latter appearing to be relatively prevalent. The housing quality appears to align with the middle-class nature of the neighborhood, characterized by well-maintained residences, while larger or luxury properties were absent. In the southern part of the neighborhood, some lower-quality housing was observed, including the workers' homes constructed in the 1950s. These homes were notably substandard, featuring minimal size and, in some cases, laminated roofs. The original back-to-back gardens had been repurposed, often for the construction of a shed or extensions to expand indoor living spaces. Although the area was generally tidy and well-kept, sporadic graffiti was visible, primarily related to the local football club, and a few areas appeared to be less well-maintained. Examples depicting these characteristics are provided in Figure 13.

Configurational analysis, as shown in Figure 14, reveals that the area boasts a single major through route along its southern border (mnNAChN: 1.09), with some local through routes appearing in warmer colors along the western and eastern edges where commercial activities are concentrated (NACh1200: 1.16). On the city-wide scale, this area does not exhibit particularly strong integration, although it fares somewhat better than Arnaout (mnNAInN: 1.09). The most integrated part of the area lies in its southwestern part, closest to the primary through route, Makariou Avenue, and the 'griddy' residential area to the west. Local integration is also somewhat lacking (mnNAIn1200: 1.12), mainly limited to the western and southern routes within the area, along with the lower segment of the eastern route. This explains the concentration of commercial activities on these roads, leaving the rest of the area notably serene.

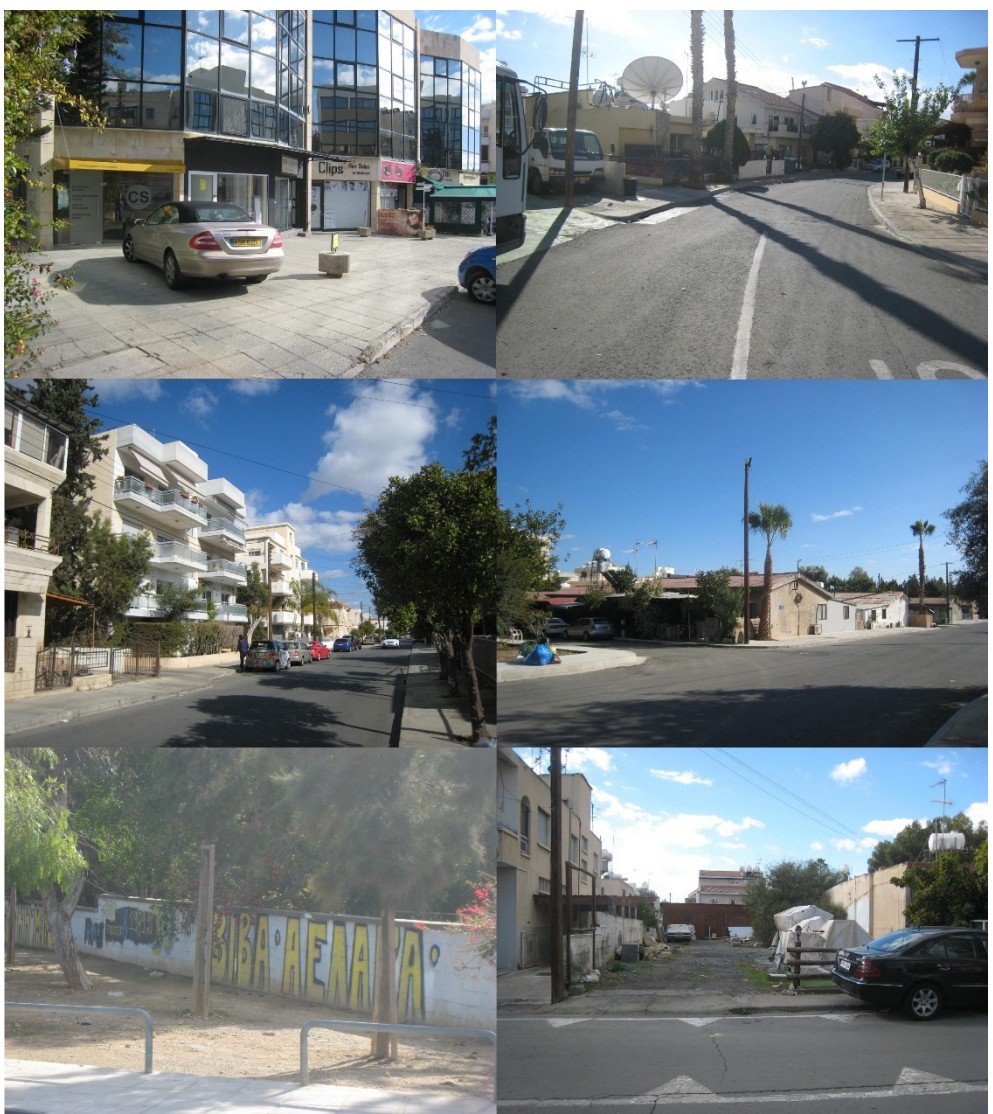

**Figure 13.** From (**left**) to (**right**) and (**top**) to (**bottom**): commercial establishments on the main road along the eastern side of the site; small, detached homes; small apartment blocks; two rows of back-to-back workers' homes with extensions filling up the back garden; graffiti related to the local football club; and an unkempt area. Source: the authors.

No archival or press sources related to this area were found, but this dearth of historical records in itself is significant. The absence of press coverage suggests that this area has been less affected by problems or controversies compared to the other two case studies. Unlike Arnaout, it does not face housing issues, nor does it encounter contentious development proposals that characterize Dasoudi, as discussed in the next section.

The primary causal factor behind the development of this area appears to be population growth. Initially, during the latter half of the 19th century, this growth led to the establishment of the cemetery. Later, in the post-World War II economic boom, the area experienced densification. Agencies operating beyond national borders played a pivotal role in initiating the pathways that led to the emergence of this area and its characteristics. The events stemming from World War II and the dissemination of social ideas that developed elsewhere influenced the construction of the city's bypass road (Makariou Avenue) and the decision to develop areas in close proximity to it, along with the construction of the workers' housing.

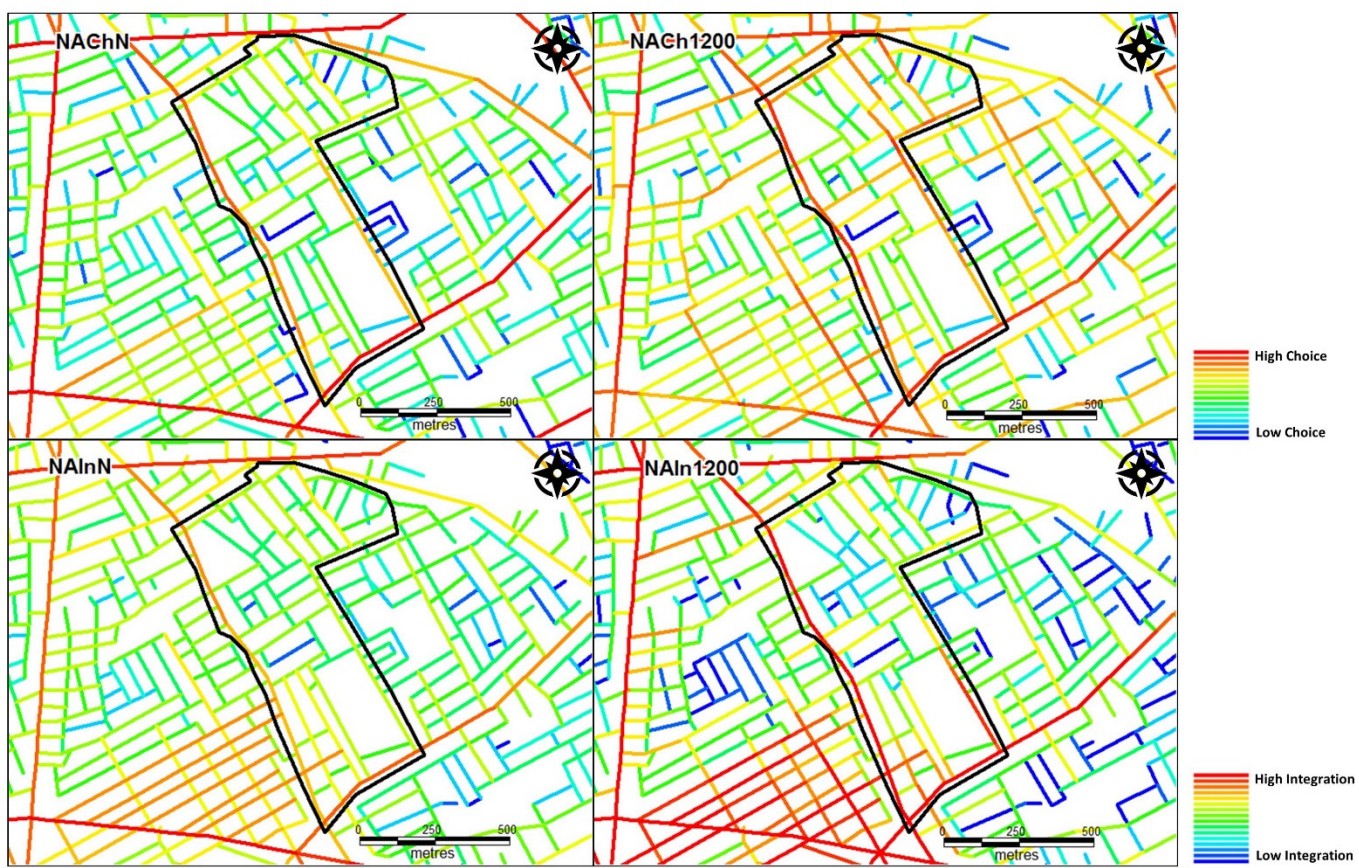

**Figure 14.** Contemporary configurational analysis of Agios Nikolaos. Source: the authors.

The choices made by developers, encompassing street layout and housing typologies, contributed to the area's distinctive character and potentially account for its higher density compared to the other two case studies. This agency, as well as that of foreign nation states, relates to the spatial configuration of the area, as a residential grid just set off from the inner ring road of the city, designed to accommodate population growth along with changing welfare ideologies. However, the spatial relationship between this area's structure and that of the entire city also plays a role in shaping its character as a tranquil residential zone. A simplified network diagram of Agios Nikolaos is presented in Figure 15.

### 3.3. Dasoudi

Dasoudi, situated on the eastern side of Limassol within the wider tourist area, literally means 'small forest', alluding to the cherished coastal woodlands beloved by local residents. These have been the subject of controversy as they were transformed into a municipal park and organized beach. The area of focus falls within the jurisdictions of the Germasogeia municipality and encompasses both the coastal zone of the park/beach managed by the Cyprus Tourism Organization (COT) and the residential sectors located to its north, with the main coastal road running between them. The study area comprises large surface area coverage but exhibits a relatively low population density with numerous voids in the built environment, including unused open spaces, undeveloped land plots, areas dedicated to small-scale agriculture, and several designated public green spaces. The area hosts a population of 2089 as per the 2011 census, but the population density is comparatively low, amounting to 2940 inhabitants per square kilometer. A substantial portion of the housing stock in this locality consists of high-rise structures, originally designed as holiday rentals, second-home properties for foreign investors, and 'retirement' homes, primarily catering to the British expatriate market. The social analysis indicates that more than 50% of the population in this area is of non-Cypriot origin, significantly surpassing the city-wide

average. Additionally, the locality boasts a notable concentration of upper-class residents and experiences average levels of unemployment. An overview of the area is shown in Figure 16.

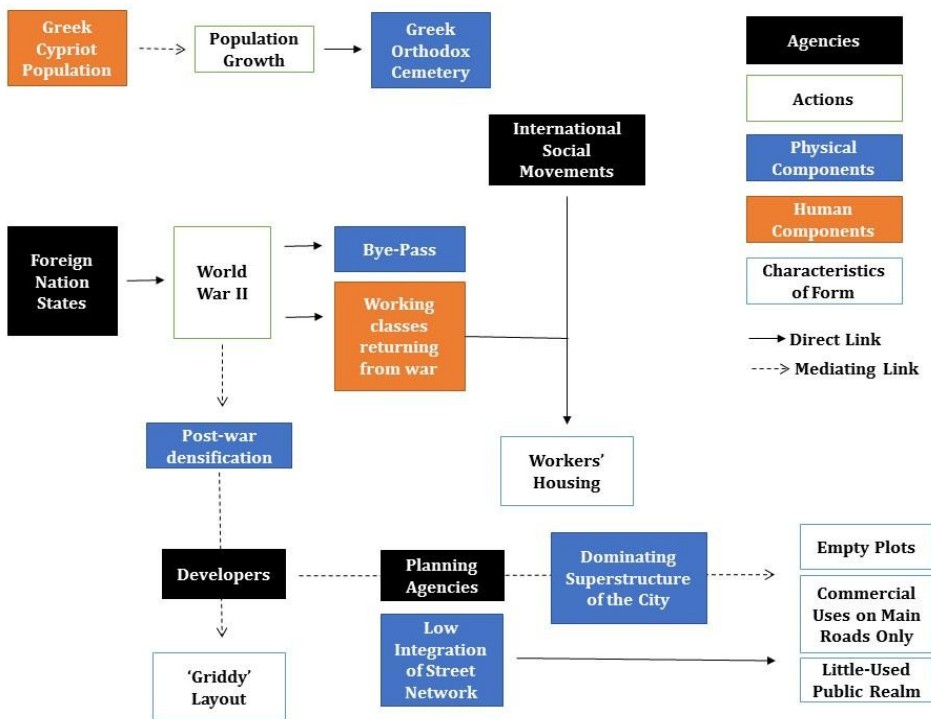

**Figure 15.** Simplified network diagram of Agios Nikolaos. Source: the authors.

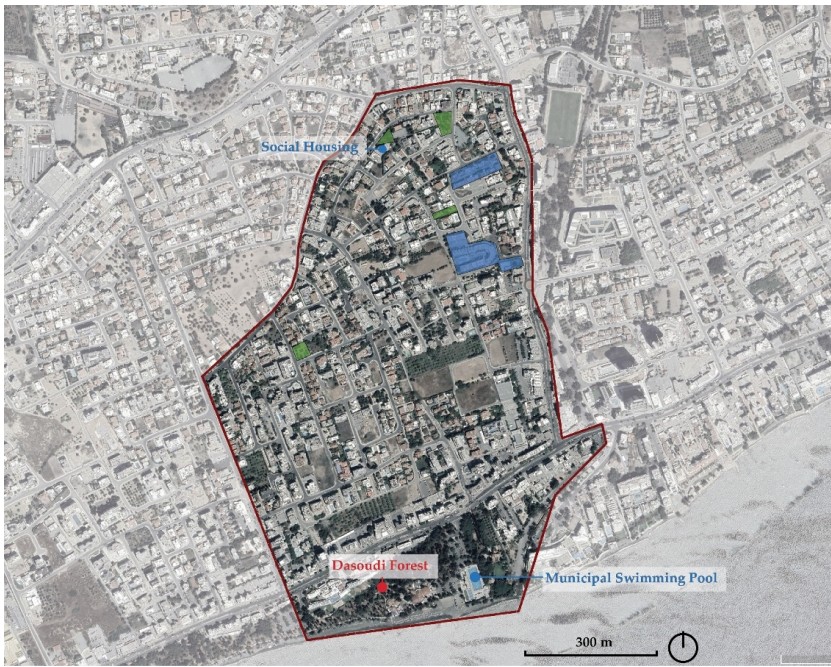

**Figure 16.** Dasoudi study area. Source: the authors; base map: Google Earth 2023.

The area comprises a small number of formal public green spaces, including play areas (highlighted in green in Figure 14), as well as two gated residential developments (highlighted in blue in Figure 14). Anecdotally, this area is known to be inhabited by wealthy Russians and retired British professionals. Although the census confirms that

upper-class residents live in this area, it is not possible to extract the nationality of non-Cypriot residents or to distinguish whether it is the Cypriot or foreign residents who belong to the upper classes. The observation and the photographic survey corroborate the anecdotal evidence in the sense that many shopping facilities are aimed at the Russian and higher-end markets and that British businesses are present in the area (Figure 17).

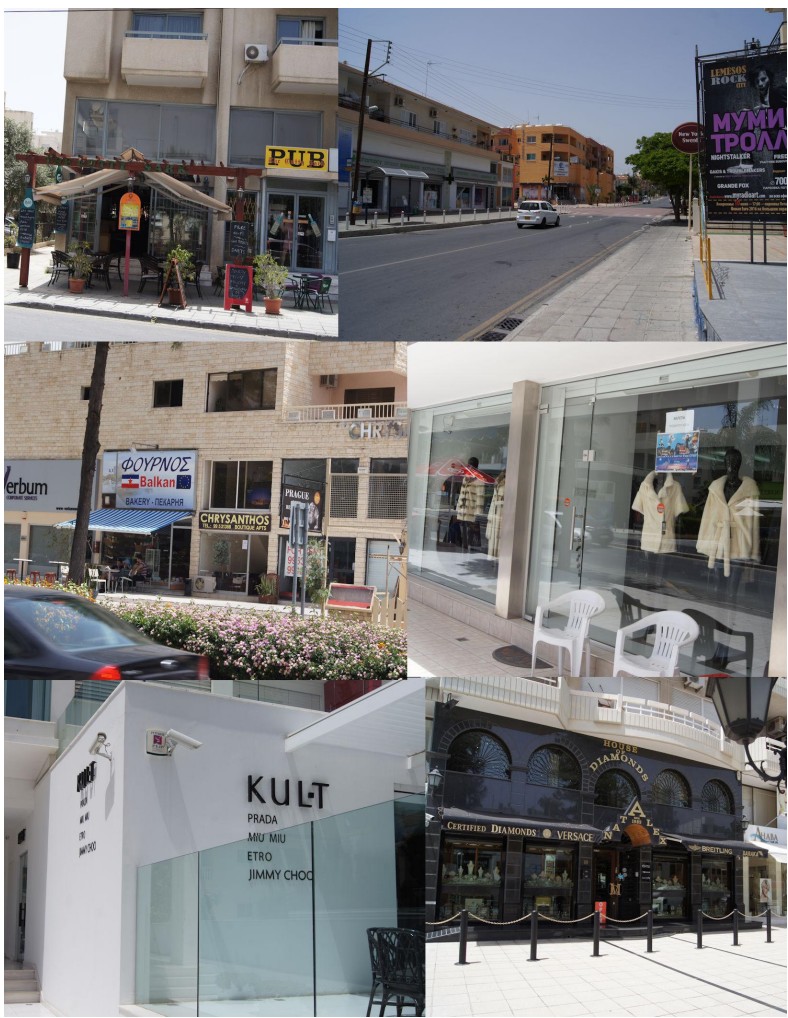

**Figure 17.** Foreign and higher-end businesses in Dasoudi (an English pub—(**top left**); pharmacy and advertising in English, Russian and Greek—(**top right**); a shop with Balkan products—(**middle left**); a fur shop—(**middle right**); and upmarket brands and jeweler's—(**bottom**)). Source: the authors.

The area was observed on a late Saturday morning in early June, and it was found to be very quiet at the time of the observation. Very few people or vehicles were observed on residential streets, with activity limited to the main roads at the edges of the area where shops and services were found. While there was much activity on the beach and park and some activity in the private gardens and swimming pools that are part of residential complexes, the public green spaces were not being used at all (Figure 18).

Configurational analysis shows that both choice and integration values are low; integration is particularly low (mnNAChN: 1.01; mnNACh1200: 1.1; mnNAInN: 0.85; mnNAIn1200: 0.81). As Figure 19 shows, the coastal road is the only long-rage route within the area, but it does not perform particularly well on the local scale: at this level, the route along the north-western side of the site has stronger local properties. The site is also highly segregated, both at the local and global level, despite the fact that, while being at the eastern end of the city, this is by no means an edge area. The very low integration levels explain the fact that this area is extremely quiet.

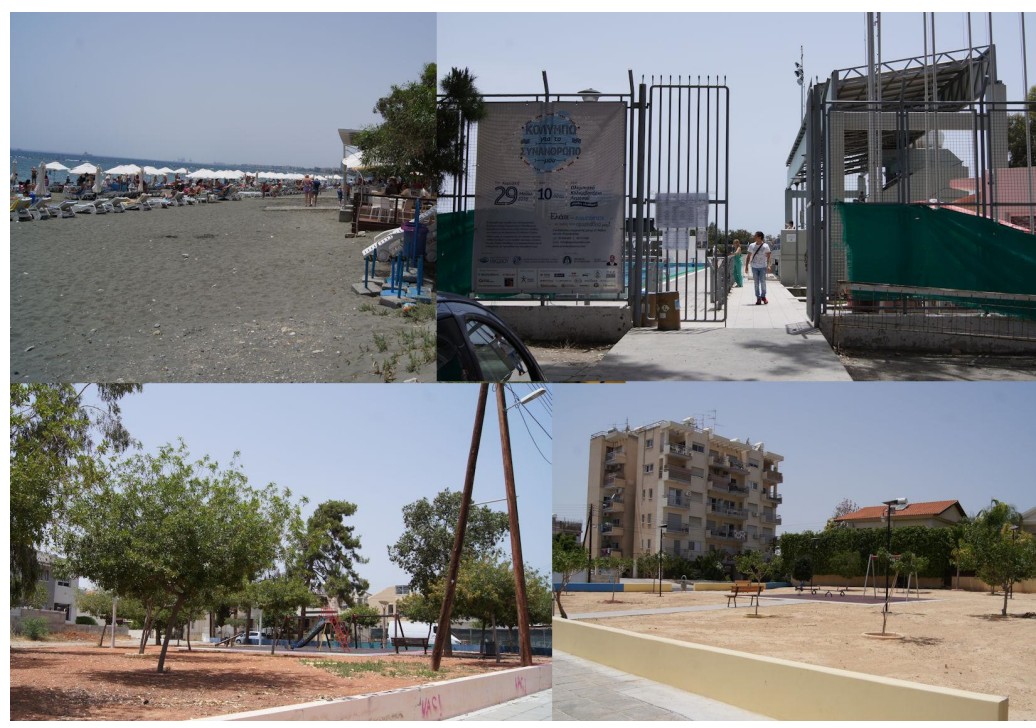

**Figure 18.** The public beach (**top left**), municipal swimming pool (**top right**) and two public spaces with play areas (**bottom**) in the residential zones of Dasoudi. Source: the authors.

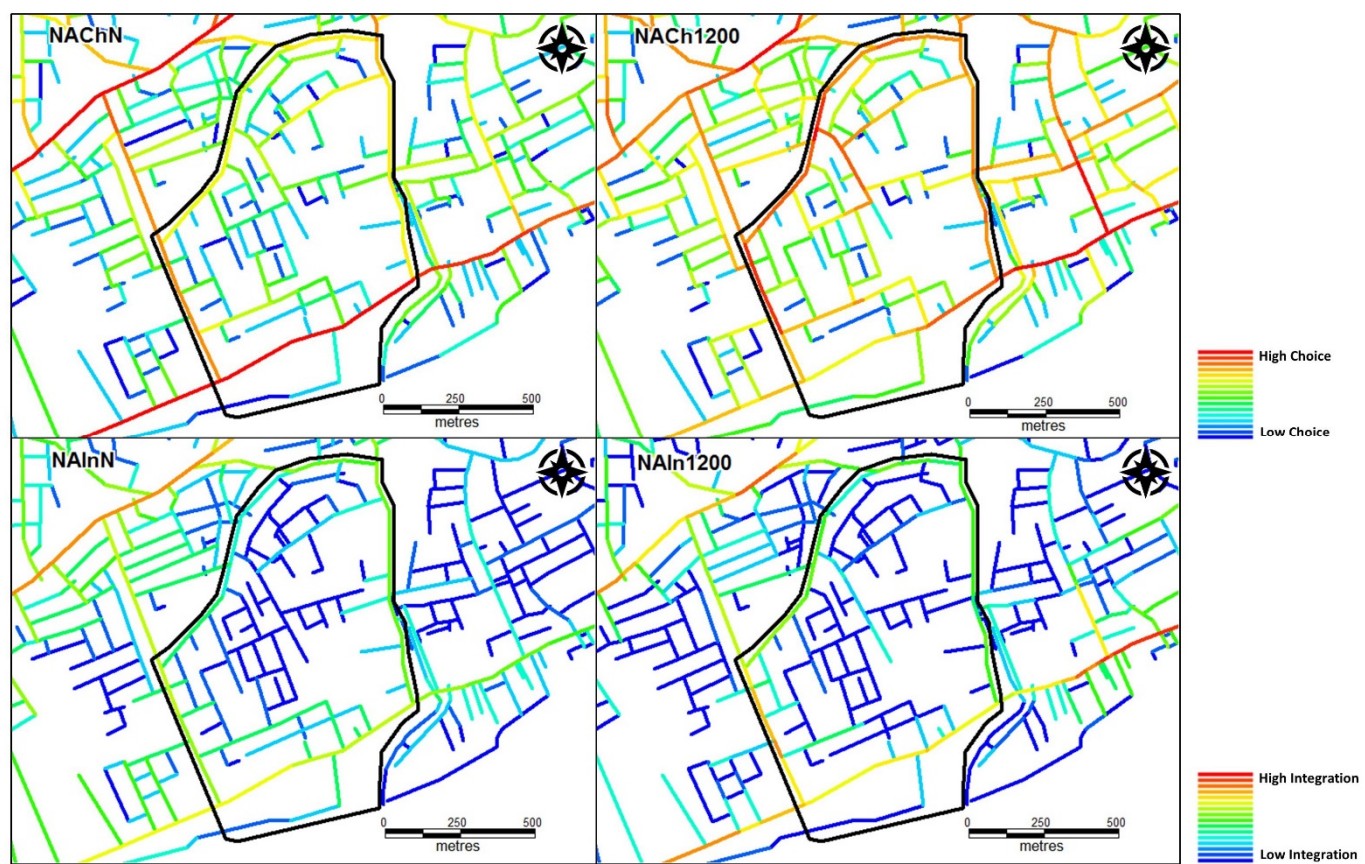

**Figure 19.** Contemporary configurational analysis of Dasoudi. Source: the authors.

The housing landscape in this area exhibits a distinctive and diverse character. While a significant portion of the residential architecture comprises high-rise apartment blocks, detached single-family homes are also present, particularly in the northern part. Among these, there is a mix of high-end villa developments and older houses. Some of these older homes feature small agricultural local holdings, possibly linked to the Greek Cypriot population that settled in the vicinity of the city either before or during the early phases of this area's extensive development. Furthermore, there are a number of apartment blocks of poorer quality, which appear to be social or refugee housing, likely built in the late 1970s or early 1980s. The landscape is punctuated by advertising for luxurious properties, reflecting a prevailing trend in the area. Notably, there are gated developments comprising both apartments and housing, some of which, despite being clearly marked as private properties, are relatively accessible.

The enduring presence of areas dedicated to agriculture is a characteristic common to many peripheral areas in Cypriot cities. Despite the long-standing existence of properties in this area and its transformation into a tourist destination since the 1970s, the urban layout maintains a low-density profile. Numerous ongoing development projects are scattered throughout the area. The diversity of housing types and land usage is showcased in Figure 20.

The development of Dasoudi is associated with the aftermath of the 1974 war, as there was a strategic effort to harness its potential for tourism after losing major tourist destinations in the northern part of the island. Nevertheless, even before 1974, the area had witnessed some residential development, with sparse housing and basic road infrastructure.

In 1987, the area remained relatively low in density, characterized by open fields, and this situation had only slightly changed by 2003. Much undeveloped land still exists, and many construction projects are ongoing. Archival research revealed that the tourism industry, particularly the Hotels Network, had set its sights on this area in the early 1970s. Discussions and debates among various stakeholders about proposals for the area's tourist development and funding occurred between 1970 and 1972 [49,51,52]. The debate revolved around who should oversee the area, whether certain sections should be privatized or remain in public hands, and questions about environmental conservation regarding the forest and the beach.

Following 1974, the development of Dasoudi for tourism became an economic imperative, leading the municipality to 'subcontract' the area in 1976 to the CTO for a nominal rent of CYP 1. The CTO subsequently generated controversial development plans [53] and released portions of the area for private use by hotels and other organizations, reaping significant profits.

In the mid-1990s, suggestions emerged for the municipality to regain control of the area [54] to counteract privatization, safeguard the environment, and better manage the locality. This move was met with legal challenges from the CTO: the management of the area and the CTO's future development plans continue to be closely examined and criticized to this day [55,56].

A wide array of groups participate in the ongoing debates surrounding Dasoudi, including political parties like AKEL (the Progressive Party of the Working People) and EDEK (the Movement for Social Democracy) alongside professional organizations such as ETEK (Cyprus Scientific and Technical Chamber), civic organizations like The Association for the Protection of Natural Environment, and business interest groups, such as the Hotels Network. The debate has seen the increasing involvement of various stakeholders and individuals, in some cases culminating in their participation in local elections to exert greater influence [57]. Plans from the early 2010s were placed on hold, and there have recently been suggestions to hold an architectural competition [58] and for improvement to consider a zero-growth model [59].

The persistent debate about the area, particularly when considering its diverse resident demographics, not only underscores the multitude of actors that can shape an area's development but also highlights its popularity among both residents and visitors. While

plans for the future management of the beach and its forest are still pending, the residential area beyond the coastline remains in a state of flux, characterized by a mix of residents from various backgrounds, including high-end permanent and vacation properties, Cypriot suburban homes intermingled with vacant lots and agricultural land, some social housing, and underutilized public spaces.

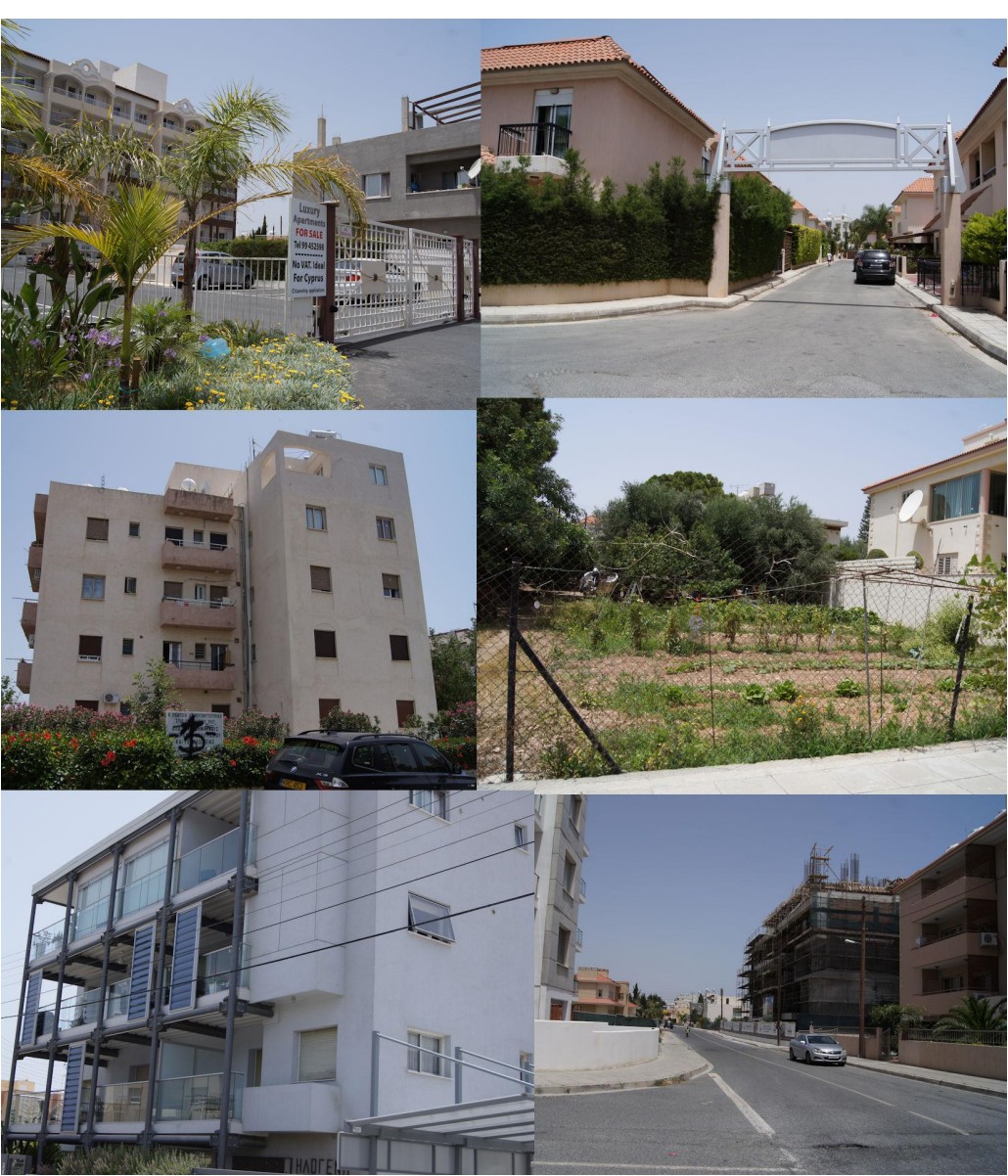

**Figure 20.** Two different types of gated developments (**top**), lower-quality housing, possibly social housing (**middle left**), a detached home with a small agricultural plot (**middle right**), a higher-end apartment block (**bottom left**), and apartments still under construction near the coastal road (**bottom right**). Source: Authors.

A simplified network diagram (Figure 21) of the area reveals that the primary force behind the emergence of this area transcends national borders. Nevertheless, the deployment of physical components and their attributes is substantially mediated by local agencies and the interactions among the human and physical elements themselves. As the area is still, to a large extent, under development, the ultimate impact of any agency on the spatial configuration still needs to stabilize; at present, the resulting effect of the agency exerted by supra-national and local entities is that of a highly segregated area. The diagram also

highlights how the spatial relationality between the area and the whole city leads to its nature as a quiet residential area and a little-used public realm beyond the major roads.

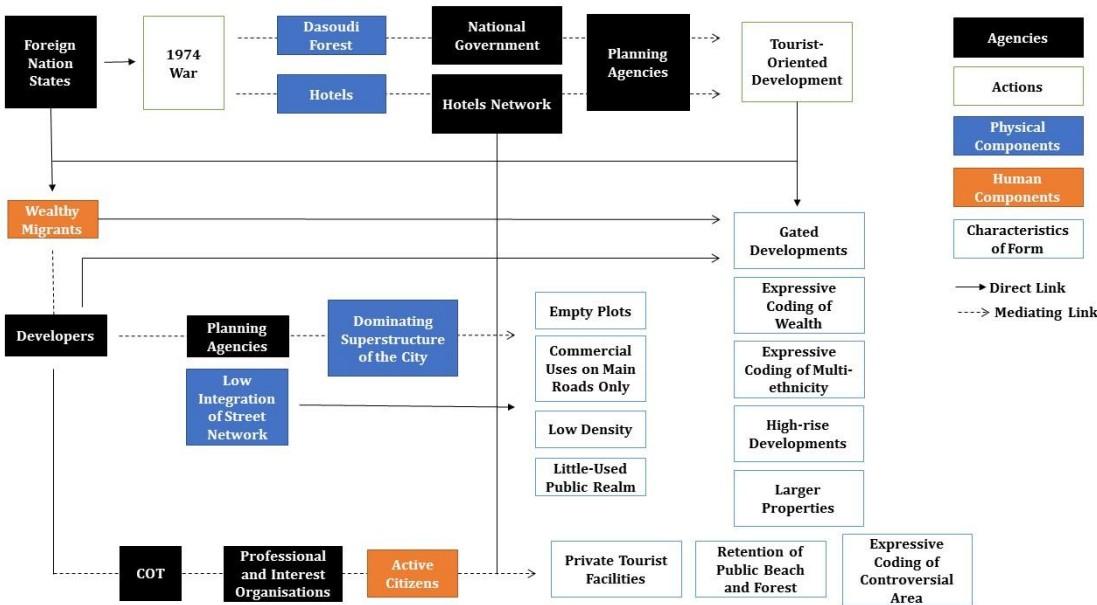

**Figure 21.** Simplified network diagram of Dasoudi as a social assemblage. Source: the authors.

## 4. Discussion

The analysis of the case study areas provides a nuanced perspective on their diversity and distinctive characteristics. Additionally, it sheds light on trends and patterns in the development of the city, dispelling potential assumptions about the relationship between these areas and the city as a whole, as well as between housing types and urban characteristics.

One notable trend is the gradual increase in building height over time. The more traditional areas primarily feature one and two-story housing, whereas the neighborhoods developed between the 1950s and 1970s exhibit a higher prevalence of two and three-story apartment buildings. Conversely, the recently developed Dasoudi area boasts a significant number of three-story and taller residential units. It is worth noting that the latter two areas also include detached homes, particularly Dasoudi, where larger properties are prevalent. These data challenge the assumption that an increase in building height directly correlates with larger block sizes and higher population densities. Instead, it underscores the significance of the overall development of fabric and density in determining population density.

Comparing the areas, Arnaout stands out with a lower population density ($5256/\text{km}^2$) in contrast to Agios Nikolaos ($6506/\text{km}^2$), despite both encompassing large blocks within the inner, earliest fringe belt of the city. Agios Nikolaos has the highest population density, even though it contains a greater proportion of vacant or temporary properties (15%) compared to Arnaout (12%). This 'anomaly' could be attributed to properties in Arnaout hosting a larger number of people, or conversely, a higher number of temporary residents might be residing within Agios Nikolaos. Meanwhile, Dasoudi exhibits an exceptionally low population density ($2940/\text{km}^2$), primarily due to its sparsely built environment but also because a substantial number of properties remain vacant or temporary (40.6%). This is potentially due to the recent nature of these developments, with many properties still awaiting sale or rental as of the 2011 census.

The physical consolidation of these areas, as depicted in Figure 22, reflects the social, economic, and political history of the island within a global context, aligned with the historical periods and events that have defined the emergence and transformation of the neighborhoods.

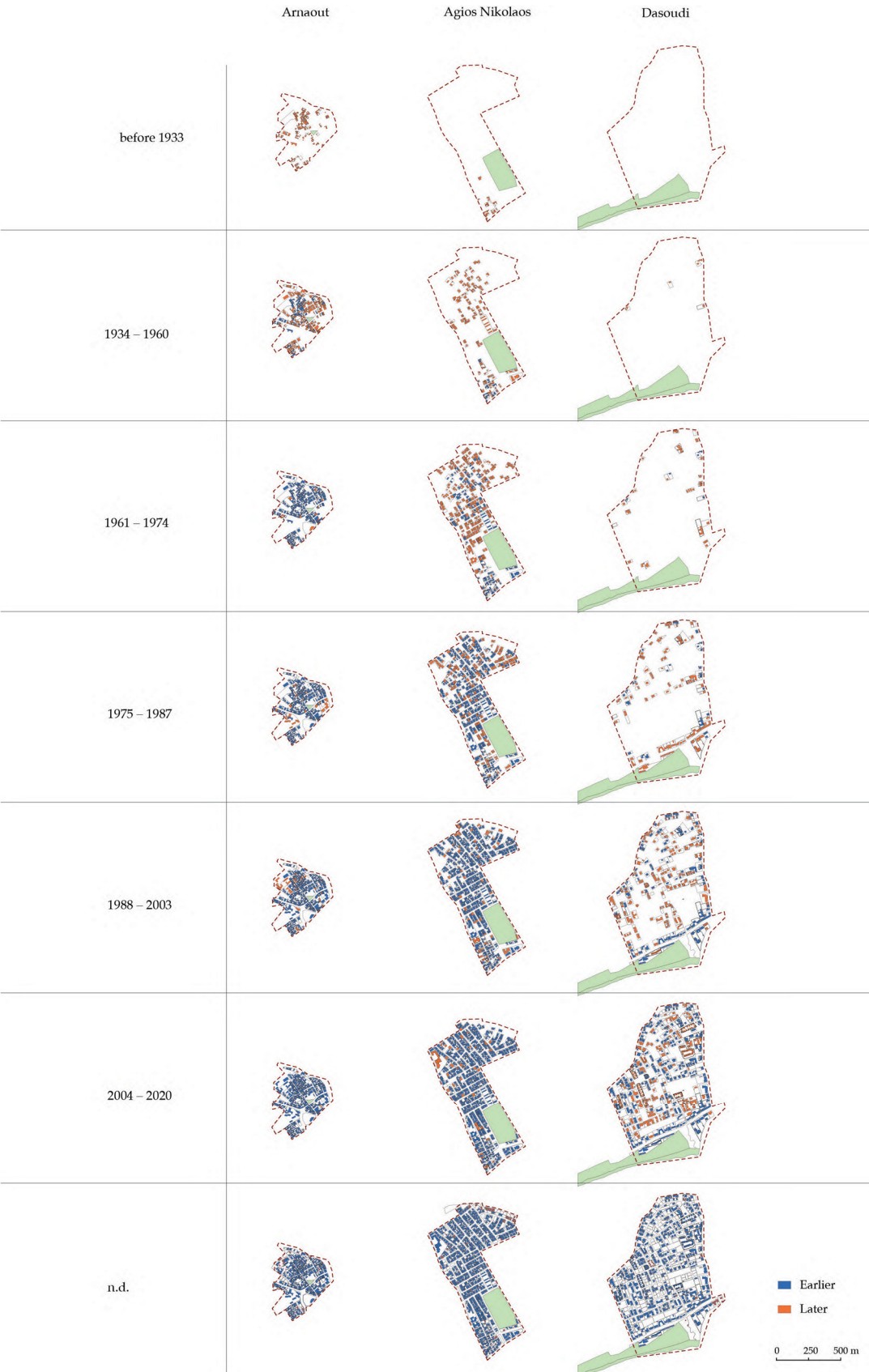

**Figure 22.** The physical consolidation of the areas. Source: the authors.

The spatial and block size values for each area across time, detailed in Table 3, also give an indication of the consolidation of the neighborhoods from a spatial and physical point of view.

**Table 3.** Summary characteristics of case study areas.

| | 1883 | 1933 | 1960 | 1974 | 1987 | 2003 | 2014 |
|---|---|---|---|---|---|---|---|
| Case Study | mnNAChN | mnNAChN | mnNAChN | mnNAChN | mnNAChN | mnNAChN | mnNAChN |
| Arnaout | 1.02 | 1.16 | 1.16 | 1.12 | 1.12 | 1.08 | 1.09 |
| Agios Nikolaos | NA | 1.12 | 1.12 | 1.10 | 1.08 | 1.09 | 1.09 |
| Dasoudi | NA | NA | NA | 1.04 | 1.00 | 0.98 | 1.01 |
| Case Study | mnNACh1200 | mnNACh1200 | mnNACh1200 | mnNACh1200 | mnNACh1200 | mnNACh1200 | mnNACh1200 |
| Arnaout | 0.95 | 1.09 | 1.13 | 1.13 | 1.14 | 1.13 | 1.14 |
| Agios Nikolaos | NA | 1.09 | 1.16 | 1.16 | 1.15 | 1.15 | 1.16 |
| Dasoudi | NA | NA | NA | 1.13 | 1.10 | 1.07 | 1.10 |
| Case Study | mnNAInN | mnNAInN | mnNAInN | mnNAInN | mnNAInN | mnNAInN | mnNAInN |
| Arnaout | 0.65 | 1.10 | 1.18 | 1.17 | 1.08 | 1.03 | 1.08 |
| Agios Nikolaos | NA | 0.87 | 1.04 | 1.04 | 0.97 | 1.08 | 1.09 |
| Dasoudi | NA | NA | NA | 0.86 | 0.78 | 0.81 | 0.85 |
| Case Study | mnNAIn1200 | mnNAIn1200 | mnNAIn1200 | mnNAIn1200 | mnNAIn1200 | mnNAIn1200 | mnNAIn1200 |
| Arnaout | 0.70 | 1.13 | 1.22 | 1.29 | 1.29 | 1.16 | 1.23 |
| Agios Nikolaos | NA | 0.96 | 1.03 | 1.13 | 1.06 | 1.08 | 1.12 |
| Dasoudi | NA | NA | NA | 0.77 | 0.75 | 0.72 | 0.81 |
| Case Study | mnBlockSize $M^2$ | mnBlockSize $M^2$ | mnBlockSize $M^2$ | mnBlockSize $M^2$ | mnBlockSize $M^2$ | mnBlockSize $M^2$ | mnBlockSize $M^2$ |
| Arnaout | NA | 17,495 | 15,835 | 16,448 | 11,713 | 10,314 | 11,528 |
| Agios Nikolaos | NA | NA | 8538 | 7759 | 7994 | 7779 | 7529 |
| Dasoudi | NA | NA | NA | 543,558 | 110,939 | 35,115 | 20,932 |

The comparative trend of the neighborhoods for each measure is displayed in Figure 23. In the case of the older neighborhoods, both integration and choice tended to increase significantly during the first three time periods in the 'life' of each neighborhood; values then tended to stabilize with slight decreases (especially in global values) from the 1980s as the city expanded significantly. In the case of the much more recent neighborhood of Dasoudi, despite the land parcellation shown by the decreasing block size since its inception in the 1970s, the pattern is different as values have not increased substantially. This indicates that consolidation has not yet taken place[7]. This finding is aligned with a previous study of the persistence and changes in spatial properties in the center and peripheral areas of Limassol [36], which highlighted how the spatial values of two high streets from different periods and in different areas of the city stabilized following their initial development period, but changes started occurring again in the 2010s. It also partially fits in with an early syntactical study of Limassol from the 1980s [60], which highlighted that the relations between local areas and the whole city were unbalanced and skewed towards a global system at the expense of local neighborhoods. However, it challenges the author's suggestion that this relationship is likely to be transitional while the urbanization of the city intensifies and stabilizes. This does not seem to have taken place, given the mean spatial values of the neighborhoods and the observed levels of activity in public spaces.

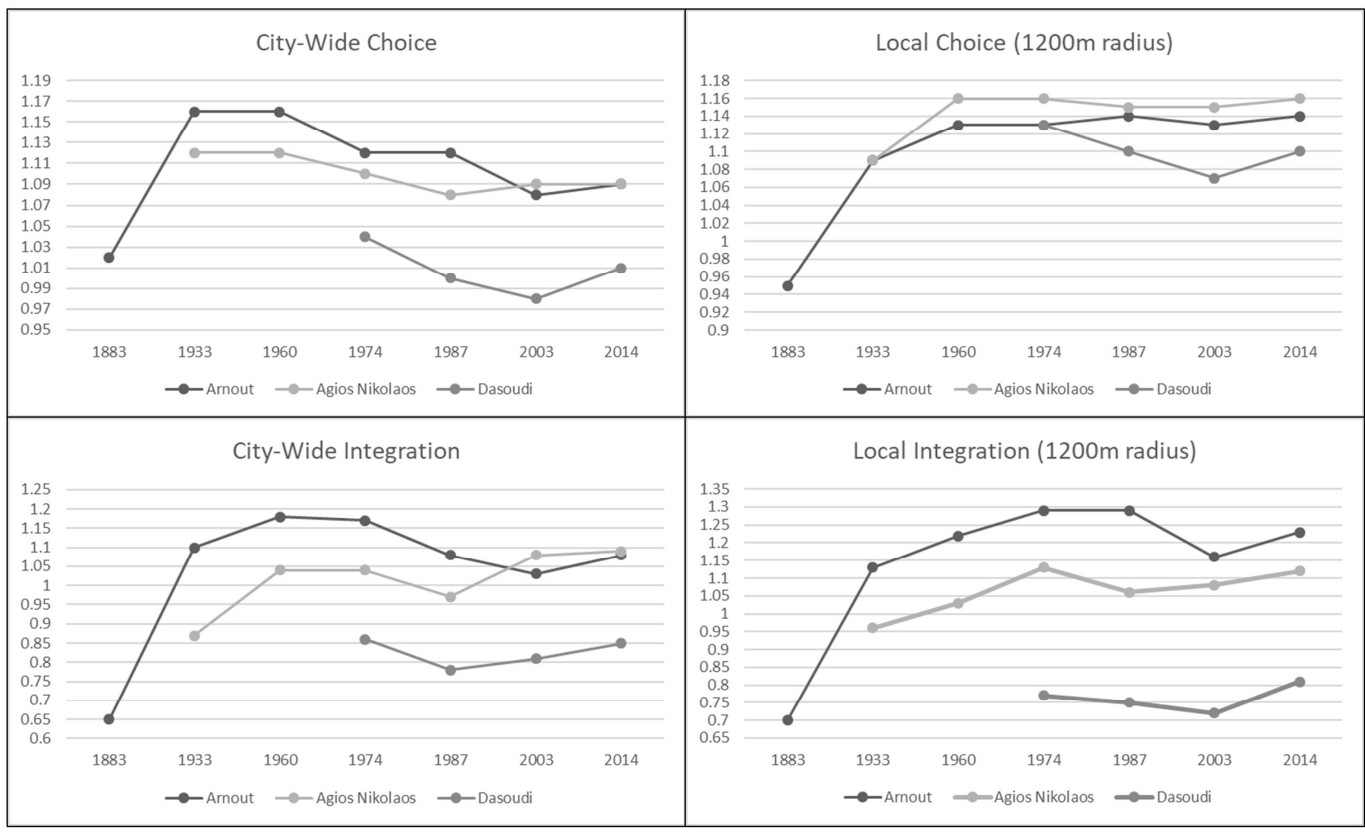

**Figure 23.** Comparative line charts of spatial values across time. Source: the authors.

Looking specifically at contemporary values, Arnaout and Agios Nikolaos exhibit somewhat similar values, with the former having stronger local values, especially for integration. These values reflect the observations of space use activities well, as Arnaout displays higher levels of movement and static activities on its most integrated streets, while Dasoudi shows almost no activity, potentially reinforced by its low population density.

These findings challenge the assumption that geographical location alone determines an area's integration within the city. They emphasize that the presence of a single highly used long-range route does not significantly impact overall accessibility values. Instead, the presence of multiple routes with high accessibility values, especially if they cross the areas rather than just bordering them, plays a more pivotal role in determining average choice and integration values.

Another significant finding is the clear and significant relationship between block size and population density. Although the focus is only on three cases, this relationship extends across the whole city [37]. However, it is essential to recognize that population density can remain high even when the average population block size is relatively large, as exemplified by Arnaout if the housing typology allows for such density.

Additionally, all areas seem to boast a considerable amount of well-maintained public green spaces. However, these spaces appear underutilized, with the notable exception of the expansive Dasoudi area. Conversely, static activities tend to gravitate toward informal or private meeting spaces—as in the example of Arnaout, where people congregate within or next to local clubs—for example, using the sidewalk as a meeting space (Figure 6). Further observations are required to truly gauge the extent of public space utilization and understand the potential reasons for underutilization. This observation challenges the prevailing perception that local neighborhoods lack green spaces. In fact, these spaces are fairly abundant. Other factors, such as their location, size, or accessibility, might contribute to the perception that they are scarce.

Whilst the research provided a wealth of findings, several limitations challenge drawing definite conclusions about some aspects of these neighborhoods' identities. Firstly, the lack of formal, quantitative observations of space use within the case studies and the broader city makes it impossible to exactly infer the relationship between the spatial properties or other physical factors and the levels of activity in public spaces. This is particularly important in this context, where pedestrian movement is known to be low throughout the urban environment and particularly in residential areas. Secondly, the data on building construction dates only represent the existing built fabric and do not capture what has disappeared. This limitation prevents providing a dynamic picture of change, including the stabilization and destabilization of the physical fabric over time. It might also lack detail on the level of consolidation in the neighborhoods at any point in time. Thirdly, at the time of the research, only 20% of the national newspaper archives had been digitized. This means that relevant narratives about the neighborhoods may be missing, limiting the historical context provided by newspaper sources[8]. Finally, syntactic analysis was limited to segment angular analysis because of the way the spatial models were constructed. This limitation means that important space syntax measures, such as connectivity (which can only be performed on axial models) describing the ease of reaching spaces within the neighborhoods, could not be assessed.

## 5. Conclusions

This research sheds light on how spatial properties (accessibility, measured as choice and integration) at various scales exert a more substantial influence than geographical location in shaping activity distribution. Furthermore, it underscores the need to consider physical characteristics like block size in conjunction with other factors, such as housing typology, to comprehend population density and its potential impact on movement in the street network. Thus, the complexity of the relationship between the entire city and its constituent parts, as well as the interplay between area characteristics, land use, activity, and population groups, is crucial to recognize. These complexities transcend simplistic assumptions about geographic locations or the influence of a single aspect on a single variable.

The agents identified in the process of the emergence of the neighborhoods and their transformation, such as their urban form and expressive characteristics, range from empires and national governments, international organizations, population groups, and local associations to small-scale material elements such as rivers, bridges, and roads. More specifically, the agency is reflected and highlighted in related demographic changes, the influx of internal and external migrant populations from differing socio-economic backgrounds, cultural identities, and housing typologies. In complex cases like the urban environment, an extensive list of all agents, actors, and components, or details about every potential capacity, would neither be possible nor valuable. Still, a greater understanding of this complexity is possible by exploring and visualizing how different components are linked together and how their capacities, as well as their actions and mediating roles, lead to the emergence of form. The diagrams do not depict a specific state of affairs highlighting permanence or even a temporal sequence of events and development, but rather a process ensuing from the interaction between human and non-human actors, from the agency exerted by national governments and international affairs and by specific small-scale physical elements of the urban fabric. The view provided by the diagrams emphasizes the neighborhood as a "contingent, fragmentary, heterogeneous but persistent product of human and non-human actors, intermediary and mediator roles, concrete associations, stabilizing and destabilizing agencies, and urban assemblages" [27] (p. 155).

Moreover, characterizing the neighborhoods as social entities and visualizing them as network diagrams provides a simplified but clear depiction of the causal factors that initiate the emergence of these areas. The analysis also elucidates the causal pathways leading from the initial establishment of physical components within the area to their expansion and development of a changing but distinct identity. The key original outcome of this study

is that the varying complexity of these diagrams across different case studies reflects the intricate nature of these entities and the differing levels of stability that they establish; this includes more complex diagrams indicating neighborhoods that have undergone or are more likely to undergo greater destabilization compared to others, due to the multiplicity of agencies involved and their potential conflicting interests. The application of the proposed framework, therefore, enables—through the diagrams—the identification of the socio-economic resilience potential of each neighborhood. The implication of this finding is that, based on the complexity of their diagram, neighborhoods may require more or less attention and sensitivity in terms of urban planning and governance, especially when social, economic, or environmental circumstances posing a threat to their stability arise at any scale (internally within the neighborhoods, within the city or in relation to national and international trends).

Looking ahead, future explorations on the topic of agency in neighborhoods should steer towards further understanding the temporal aspect of the stabilization of form, the weight of different factors in the potential socio-economic resilience of places, and how these findings, particularly the output of diagrams with their related inferences, can be translated into practical planning and governance actions.

**Supplementary Materials:** The following supporting information can be downloaded at: https://www.mdpi.com/article/10.3390/land13030269/s1. Vector layer S1: georeferenced processed space syntax models of Limassol 1883, 1933, 1960, 1974, 1987, 2003, 2014 (tab file format); Vector layer S2: georeferenced block size map of Limassol 1883, 1933, 1960, 1974, 1987, 2003, 2014 (tab file format).

**Author Contributions:** Conceptualization, I.G., N.C. and C.C.; methodology, I.G.; formal analysis, I.G.; investigation, I.G.; data curation, I.G. and A.R.; writing—original draft preparation, I.G.; writing—review and editing, N.C., C.C., I.G. and A.R.; visualization, I.G. and A.R.; supervision, N.C. and C.C.; project administration, I.G. All authors have read and agreed to the published version of the manuscript.

**Funding:** This research received no external funding.

**Data Availability Statement:** Data are contained within the supplementary material with the exception of third party data (building construction dates)—commercial IPR restrictions apply to these data; these data were obtained from Ask Wire.

**Acknowledgments:** The authors are grateful to the company Ask Wire for providing the data of building construction dates visualized in Figure 22.

**Conflicts of Interest:** The authors declare no conflicts of interest.

## Notes

[1]  Here, 'agency' is defined as the capacity to exert an influence on something.

[2]  Six conversations, in the form of semi-structured interviews, were held with local experts regarding the development of Limassol as part of a broader study; out of these, three (two former municipal planning officers and one historian) mentioned the case studies selected for this research.

[3]  Space syntax is both a theory and a method for quantitatively describing patterns of spatial layout and relating these patterns to social activities such as movement, behavior, and even social meaning and interpretation. Space syntax theory is based on the fundamental idea that space is an intrinsic aspect of human activity and that the social effects of urban space ensue from how the inter-relations between spaces combine to form a city as a whole [42]. Space syntax measures correlate with a variety of socio-economic characteristics in urban environments and, specifically, are representative of movement at different scales [43]. Within the scope of this paper, city-wide scales provide a description of how these neighborhoods are related to the city.

[4]  The analysis used in this study is angular segment analysis, which takes into account the least angular deviation of each street segment from all other segments. For further details of space syntax theory and methodology, see www.spacesyntax.org (accessed on 13 February 2024), along with Hillier and Hanson [42], Hillier [45], and van Nes and Yamu [46]. For details of angular analysis, see Hillier and Iida [44]. The spatial model was created based on the OpenStreetMap road center line, which was manually edited against contemporary Google and Bing map and satellite data in MapInfo software version 8.5. The analysis was performed using the DepthMap Process tool in the space syntax extension for MapInfo, developed and licensed by Space Syntax Ltd.

<sup></sup>5    The data presented certain challenges. Approximately 5–10% of the building information had gaps, with a notably higher rate in the case of Arnaout. To address these gaps, the authors manually filled in missing data, often relying on assumptions drawn from historical maps for construction dates. These same historical maps were used for the diachronic space syntax analysis of the whole city in a broader study [38], providing the contextual information for the present research.

6    Specifically, the photographic survey took place on 19 December 2015 (Arnaout), 15 December 2016 (Agios Nikolaos), and 4 June 2016 (Dasoudi).

7    It should, of course, be noted that the first four periods of analysis (1883–1974) span a much longer time than the last four (1974–2014). Furthermore, the area is relatively peripheral in the spatial models, and an edge effect may impact the values as the extent of the historical models is dictated by the extent of the available historical cartography. However, this effect is partly 'natural' since the area is delimited to the east by the Germasogia River, with the only crossing point until 1987 being the coastal road. The model for 2014 is based on digital cartography and, therefore, includes the area to the east of Dasoudi, including a new connection over the river just to the north of the area; the values all show a slight increase at this point.

8    The 20% coverage of the digital archives was extensive for major newspapers and more restricted for other newspapers; it also focused on the years of significant historical events, such as the World Wars, the establishment of the Republic of Cyprus, and the 1974 war.

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
