# Peer review of "Agency within Neighborhoods: Multi-Scalar Relations between Urban Form and Social Actors"

_land, doi:10.3390/land13030269_

Round 1

Reviewer 1 Report (Previous Reviewer 3)

Comments and Suggestions for Authors

As I mentioned before, the article is very interesting and it was really pleasant to read. The author's have addressed all my previous suggestions.

As I was reading one last time the article I would think it would be interesting, if the author's agree, to talk a little bit about urban pleasantness how it can affect the urban environment as it can help to understand the formation of neighborhoods.

For example the percentage of green areas and building high can provide part of the explanation to why some neighborhoods grow and become more attractive to future and current residents, while others, do not.

Again, I'll leave this to the author's to evaluate if the article should, or not, include just a small paragraph address this idea and how it can correlate with the work presented.

I'll leave here a few references:

https://doi.org/10.1080/13574809.2018.1554994

https://spaj.ukm.my/jsb/index.php/jbp/article/view/170

https://doi.org/10.3390/land12040878

https://www.semanticscholar.org/paper/Measuring-Beauty-in-Urban-Settings-Calafiore/c5a631d927797461bd663efacef08431032c0687

https://doi.org/10.1016/0169-2046(94)90020-5

https://doi.org/10.1177/23998083211056341

Comments on the Quality of English Language

Minor editing of English language required

Author Response

Thank you for the suggestion and the references provided. We agree with your comment, but have decided to leave out this aspect since we would be unable to provide any metric regarding urban pleasantness to support such a discussion even if brief. However, it is certainly something we will consider expanding on for future research.

Some critical points from other reviews had to be addressed and the article is increasing substantially in length, so we also want to avoid making it even longer unless an addition is strictly necessary.

Reviewer 2 Report (Previous Reviewer 2)

Comments and Suggestions for Authors

Although substantial revisions have been made to enhance the paper's quality, there are still some parts that authors could not address adequately. Consequently, this paper lacks sufficient qualifications. Please bear in mind that the issue of plagiarism is often misunderstood by researchers. You can easily obtain a plagiarism checker report by rephrasing your words, which, in my opinion, does not meet the criteria for sound research. All novel statements must be supported by evidence. Surprisingly, the authors made such claims in the Introduction without sufficient evidence and citations. In the Results section, where novel statements should be supported, they instead used resources from other works. This is why they could not address the paper's shortcomings professionally. Please note that while your work may be considered the best paper in the eyes of other reviewers, in my opinion, it lacks the standard qualifications of a scientific paper.

Kindly refer to the highlighted parts of the uploaded PDF.  It must be noted that authors have to provide reliable and authentic reference and evidence for all the highlighted points.

Author Response

First of all, let us apologize for not addressing appropriately the parts highlighted in the PDF in the previous revision. We are unsure if the document had been provided then and this time we had to inquire with the editor about the sections you were referring to and obtained the PDF.

We have kept the relevant text highlighted in this revision, with edited text and additional references in red. We have addressed the comments / highlights as follows:

  • A small amount of rephrasing and addition of text for clarification of our statements
  • Provided external references to support the statements and to acknowledge previous research where, indeed, other scholars had already made such statements
  • Provided quotations with references where our statements where explanations / paraphrasing of original theories (e.g. ANT)
  • Clarified when a statement is based solely on our previous research, providing further references. We fully understand that this is self-referencing occurring in one case (line 182-184). We have done substantial research in relation to this topic over almost ten years, so we hope that this is appropriate, but please, let us know if you feel it is not.

Regarding the Results section, the work illustrated was carried out within the specific framework set out in the Introduction. The analysis (spatial history / narrative) is presented in a somewhat ‘stand-alone’ format. This is to then assess in the Discussion whether the analysis and output of the diagrams ensuing from the application of the framework is valuable in revealing how agency is reflected in the physical form and expressive features of neighborhood. Here the results are elaborated with the addition of an explorative assessment of the physical consolidation of neighborhoods and the diachronic trend of spatial configuration values for the three neighborhoods.

We may be misunderstanding what you mean and are somewhat unsure about the ‘novel statements’ you are referring to and how we are using ‘resources from other works’. We would be grateful if you wish to clarify.

Reviewer 3 Report (Previous Reviewer 1)

Comments and Suggestions for Authors

Thank you for your efforts and enormous works of improving this research.

One thing that I’d like to mention about the process is the simplified network diagrams. There are three diagrams with some significant agencies and direct or indirect relationships to the characteristics of the selected neighborhoods, but you didn’t mention about how these agencies relate to the features of spatial configurations. It means that when you explain these important agencies (figure 9, 15, and 21) with the comparative line charts (figure 23) in the section of discussion, it would be helpful to understand the relationship between agencies and urban morphologies. 

Use we instead of authors...

Comments on the Quality of English Language

Thank you for your efforts and enormous works of improving this research.

One thing that I’d like to mention about the process is the simplified network diagrams. There are three diagrams with some significant agencies and direct or indirect relationships to the characteristics of the selected neighborhoods, but you didn’t mention about how these agencies relate to the features of spatial configurations. It means that when you explain these important agencies (figure 9, 15, and 21) with the comparative line charts (figure 23) in the section of discussion, it would be helpful to understand the relationship between agencies and urban morphologies.

 Use "we" instead of "authors"

Author Response

We have added a sentence to summarize the key relationship between the agency and the spatial / physical configurations of the neighborhoods in each instance, above each network diagram.

In the previous revision we were required to change all instances of “we” to “authors”. Usage is now consistent; we will comply with editorial guidelines as directed if this needs changing.

Round 2

Reviewer 2 Report (Previous Reviewer 2)

Comments and Suggestions for Authors

It is accepted.

This manuscript is a resubmission of an earlier submission. The following is a list of the peer review reports and author responses from that submission.

Round 1

Reviewer 1 Report

Comments and Suggestions for Authors

It is pleased to take a chance to review this article, and impressed for authors to make an effort to illustrate various forms of agency within the neighborhoods and its relationship to the city.

Especially, it is excellent that authors explain the entire process from the present limitations of understanding substantial ways of how neighborhoods have been shaped at both a global scale and local one by using Assemblage Theory and Actor-Network Theory which are helpful to combine social elements, physical components, and spatial layouts, throughout different three regions in Cyprus.

However, it needs to be improved in the conclusion and the abstract sections, since it is not actually clearly explained what kinds of forms of agency would play essential roles in shaping the neighborhoods. Of course, you have mentioned some positive relationships between some agencies like block size and housing typology and the using pattern of public space. But it is hard to comprehend how it would be differentiated among three chosen cases. Especially, you have to insert some tables which are strongly related to the comment in the section of conclusion, such as line 749 (static activities tend to gravitate toward informal or private meeting spaces…). Naïve readers who are lack of information about your cities do hardly understand what it means. Also, in line 755, ‘spatial properties’ are mentioned, but it is not fully explained what kinds of properties.

You inserted three diagrams which are representative images for the cases. But you have to think about the ways of representing the diagram. I think if these diagrams are chronologically aligned, it would be good for readers to follow and understand their historical changes and some important factors which have affected the spatial structure and land uses. (there are too many information described in the diagram, but it is hard to follow)  

Thank you again for your time and efforts. I really look forward to having a look at the revised one soon. 

Reviewer 2 Report

Comments and Suggestions for Authors

I would like to express my gratitude to the authors for their work on the current manuscript. However, it is essential to address some significant drawbacks present in the manuscript. There are several shortcomings and weaknesses that require substantial revision, as outlined in the following comments.

1.       Your topic is ambiguous; what do you mean by "agency"? Its relationship with urban morphology should be better clarified. The authors should be more precise in their terminology selection.

2.       It is suggested to begin the abstract by explaining the research significance for conducting this study.

3.       The global factors mentioned in the findings part of the abstract should be exemplified.

4.       What is the original outcome of this study?

5.       The research scope must be further clarified and justified.

6.       Bibliographic resources are weak and should be strengthened with 20-30 more up-to-date citations.

7.       Further clarification is needed in the Introduction regarding terms such as 'agency,' 'human and organizational actors,' and 'global and local' factors.

8.        What are the implications of this study?

9.       There are several sentences without proper citations, which could be considered a form of plagiarism. This must be corrected.

10.   Consider converting Table 1 into a diagram for better visualization.

11.   On page 4 of 29, the first half of the page should be summarized.

12.   The authors should clearly state their research questions.

13.   The geographical location of the study areas (including country, city, neighborhoods) should be displayed on a map. Additionally, information about the socio-cultural context and the dominant religion of the study area would be useful.

14.   Explain why only three experts were consulted. Is this sample size adequate? How was it determined that this range of contributions would provide comprehensive insights?

15.   All adopted methods should be presented in a flowchart to visualize the research design.

16.   Explain why the syntactical analysis focused only on integration and choice and did not assess configurational attributes with connectivity.

17.   The definition of integration needs to be more precise and accurate.

18.   Clarify the simulator software used for space syntax analysis.

19.   The manuscript needs to be reviewed for grammatical and punctuation accuracy.

20.   Include the details of your conversations with the local experts in the appendix.

21.   Specify the season (date) and hours when your observations took place.

22.   Clearly outline the primary aims of taking photography observations and their relevance to your research.

23.   Explain the significance of displaying figures 2, 3, 4, 6, 8, 10, 11, 15, 16, and 18. Are all of these figures necessary?

24.   Discuss the extent to which the syntactical analysis aligns with the observations.

25.   Please refrain from interpreting the most integrated parts as automatically the most accessible parts. This is a relative fact, and misinterpreting is possible. If you’re insisted to allude accessibility, you can incorporate your finding with connectivity.

26.   Why is it that all the photos based on your observations lack pedestrians? It's surprising that there are no signs of social, optional, or even obligatory human activities. It seems as if no one lives in these areas. Could it be due to incorrect timing during the observations?

27.   Consider using an analytical approach in the results section instead of being overly descriptive. Using other resources in your results section lacks novelty.

28.   Summarize pages 12, 17, and 24.

29.   Syntactical analysis should be extended for all the study areas. Space syntax analysis could benefit from more inferential insights.

30.   Separate the discussion and conclusion sections.

31.   Explain why syntactical analysis is limited to contemporary configurations. Comparing the results with morphological attributes of previous periods could significantly enrich the study.

32.   Consider presenting Table 3 as a line chart.

33.   Develop suggestions for further research more thoughtfully.

34.   Findings of this study must be compared with the previously conducted studies in discussion section.

35.   What are the limitations of this study?

Comments on the Quality of English Language

It is included in the comments.

Reviewer 3 Report

Comments and Suggestions for Authors

The article was interesting to read. I have a few small annotations that can improve the quality of the article.

“The authors”; “We”. Please carefully read the text and decide if you want the article to be fully written in the third person of the singular or plural.

LINE 33: “However, our attempts to define and describe neighborhoods often oversimplify their complex social networks in order to streamline analysis and are potentially influenced by normative biases.” - I would suggest dividing this sentence into two.

Overall introduction is very complete, but, it would be interesting to address a bit more the relationship between neighborhoods and the city, i.e., their importance, their role, governance… Considering that this is the main focus of the article, it would be adequate to make a small introduction about their relationship, and naturally, referencing the needed literature that analyses their relationship.

Materials and Methods - Before presenting the neighborhoods it would be interesting just a small paragraph presenting the city of Limassol so the reader also understands where those neighborhoods are integrated. Additionally, and given the history of Cyprus, this small presentation of city could be accompanied with a map. If the author’s find it to be relevant, a small resume of the city history. If the Arnaout was originally Turkish Cypriot and now is not, it might be relevant just to mention that.

Space Syntax - Was it used previously to works similar to what is being presented? If so, please reference said works. Additionally, for those that are not fully familiar with space syntax it would be interesting to provide a small context of what it is and its capabilities (with references).

Figure 5, figure 12, figure 17: The way it was added to the article it appears to be only 1 map, instead of four. Please separate the maps plus add legend for the colours (“from red to blue” in the description is not enough) and the north arrow.
All figures (including maps of each neighborhoods) need to be revised according to this: add the necessary legends in the figures so the author can fully understand what is being shown without having to look for meaning in the text.

LINE 451: “this area lies just to the northeast of the roundabout located at the eastern end of Makariou Avenue.” - I imagine that for locals this is a very good reference point but for anyone else, this adds absolutely nothing. The best way to explain their location is to present a city map and the neigboorhood locations. The authors can add reference points to help the location. However, like it is, it does not say anything.

Comments on the Quality of English Language

Minor editing of english
